# CUTS: Neural Causal Discovery from Irregular Time-Series Data

**Yuxiao Cheng**[1]    **Runzhao Yang**[1]    **Tingxiong Xiao**[1]    **Zongren Li**[3]
**Jinli Suo**[12*]    **Kunlun He**[3*]    **Qionghai Dai**[12*]
[1]Department of Automation, Tsinghua University
[2]Institute for Brain and Cognitive Science, Tsinghua University (THUIBCS)
[3]Chinese PLA General Hospital
{cyx22,yangrz20,xtx22}@mails.tsinghua.edu.cn
{lizongren,kunlunhe}@plagh.org  {jlsuo,qhdai}@tsinghua.edu.cn

## Abstract

Causal discovery from time-series data has been a central task in machine learning. Recently, Granger causality inference is gaining momentum due to its good explainability and high compatibility with emerging deep neural networks. However, most existing methods assume structured input data and degenerate greatly when encountering data with randomly missing entries or non-uniform sampling frequencies, which hampers their applications in real scenarios. To address this issue, here we present CUTS, a neural Granger causal discovery algorithm to jointly impute unobserved data points and build causal graphs, via plugging in two mutually boosting modules in an iterative framework: (i) *Latent data prediction stage*: designs a Delayed Supervision Graph Neural Network (DSGNN) to hallucinate and register irregular data which might be of high dimension and with complex distribution; (ii) *Causal graph fitting stage*: builds a causal adjacency matrix with imputed data under sparse penalty. Experiments show that CUTS effectively infers causal graphs from irregular time-series data, with significantly superior performance to existing methods. Our approach constitutes a promising step towards applying causal discovery to real applications with non-ideal observations.

## 1 Introduction

Causal interpretation of the observed time-series data can help answer fundamental causal questions and advance scientific discoveries in various disciplines such as medical and financial fields. To enable causal reasoning and counterfactual prediction, researchers in the past decades have been dedicated to discovering causal graphs from observed time-series and made large progress (Gerhardus & Runge, 2020; Tank et al., 2022; Khanna & Tan, 2020; Wu et al., 2022; Pamfil et al., 2020; Löwe et al., 2022; Runge, 2021). This task is called causal discovery or causal structure learning, which usually formulates causal relationships as Directed Acyclic Graphs (DAGs). Among these causal discovery methods, Granger causality (Granger, 1969; Marinazzo et al., 2008) is attracting wide attentions and demonstrates advantageous due to its high explainability and compatibility with emerging deep neural networks (Tank et al., 2022; Khanna & Tan, 2020; Nauta et al., 2019)).

In spite of the progress, actually most existing causal discovery methods assume well structured time-series, i.e., completely sampled with an identical dense frequency. However, in real-world scenarios the observed time-series might suffer from **random** data missing (White et al., 2011) or be with **non-uniform periods**. The former is usually caused by sensor limitations or transmission loss, while the latter occurs when multiple sensors are of distinct sampling frequencies. Robustness to such data imperfections is urgently demanded, but has not been well explored yet so far. When confronted with unobserved data points, some straightforward solutions fill the points with zero padding, interpolation, or other imputation algorithms, such as Gaussian Process Regression or neural-network-based approaches (Cini et al., 2022; Cao et al., 2018; Luo et al., 2018). We will show in the experiments section that addressing missing entries via performing such trivial data imputation in a pre-processing manner would lead to hampered causal conclusions.

---

*Corresponding author

To push causal discovery towards real applications, we attempt to infer reliable causal graphs from irregular time-series data. Fortunately, for data that are assumed to be generated with certain causal structural models (Pamfil et al., 2020; Tank et al., 2022), a well designed neural network can fill a small proportion of missing entries decently given a plausible causal graph, which would conversely improve the causal discovery, and so forth. Leveraging this benefit, we propose to conduct causal discovery and data completion in a mutually boosting manner under an iterative framework, instead of sequential processing. Specifically, the algorithm alternates between two stages, i.e., (a) *Latent data prediction stage* that hallucinates missing entries with a delayed supervision graph neural network (DSGNN) and (b) *Causal graph fitting stage* inferring causal graphs from filled data under sparse constraint utilizing the extended nonlinear Granger Causality scheme. We name our algorithm **C**ausal discovery from irreg**U**lar **T**ime-**S**eries (CUTS), and the main contributions are listed as follows:

- We proposed CUTS, a novel framework for causal discovery from irregular time-series data, which to our best knowledge is the first to address the issues of irregular time-series in causal discovery under this paradigm. Theoretically CUTS can recover the correct causal graph with fair assumptions, as proved in Theorem 1.

- In the data imputation stage we design a deep neural network DSGNN, which successfully imputes the unobserved entries in irregular time-series data and boosts the subsequent causal discovery stage and latter iterations.

- We conduct extensive experiments to show our superior performance to state-of-the-art causal discovery methods combined with widely used data imputation methods, the advantages of mutually-boosting strategies over sequential processing, and the robustness of CUTS (in Appendix Section A.4).

## 2 RELATED WORKS

**Causal Structural Learning / Causal Discovery.** Causal Structural Learning (or Causal Discovery) is a fundamental and challenging task in the field of causality and machine learning, which can be categorized into four classes. (i) Constraint-based approaches which build causal graphs by conditional independence tests. Two most widely used algorithms are PC (Spirtes & Glymour, 1991) and Fast Causal Inference (FCI) (Spirtes et al., 2000) which is later extended by Entner & Hoyer (2010) to time-series data. Recently, Runge et al. propose PCMCI to combine the above two constraint-based algorithms with linear/nonlinear conditional independence tests (Gerhardus & Runge, 2020; Runge, 2018b) and achieve high scalability on large scale time-series data. (ii) Score-based learning algorithms based on penalized Neural Ordinary Differential Equations (Bellot et al., 2022) or acyclicity constraint (Pamfil et al., 2020). (iii) Convergent Cross Mapping (CCM) firstly proposed by Sugihara et al. (2012) that tackles the problems of nonseparable weakly connected dynamic systems by reconstructing nonlinear state space. Later, CCM is extended to situation of synchrony (Ye et al., 2015), confounding (Benkő et al., 2020) or sporadic time series (Brouwer et al., 2021). (iv) Approaches based on Additive Noise Model that infer causal graph based on additive noise assumption (Shimizu et al., 2006; Hoyer et al., 2008). Recently Hoyer et al. (2008) extend ANM to nonlinear models with almost any nonlinearities. (v) Granger causality approach proposed by Granger (1969) which has been widely used to analyze the temporal causal relationships by testing the aid of a time-series on predicting another time-series. Granger causal analysis originally assumes that linear models and the causal structures can be discovered by fitting a Vector Autoregressive (VAR) model. Later, the Granger causality idea was extended to nonlinear situations (Marinazzo et al., 2008). Thanks to its high compatibility with the emerging deep neural network, Granger causal analysis is gaining momentum and is used in our work for incorporating a neural network imputing irregular data with high complexities.

**Neural Granger Causal Discovery.** With the rapid progress and wide applications of deep Neural Networks (NNs), researchers begin to utilize RNN (or other NNs) to infer nonlinear Granger causality. Wu et al. (2022) used individual pair-wise Granger causal tests, while Tank et al. (2022) inferred Granger causality directly from component-wise NNs by enforcing sparse input layers. Building on Tank et al. (2022)'s idea, Khanna & Tan (2020) explored the possibility of inferring Granger causality with Statistical Recurrent Units (SRUs, Oliva et al. (2017)). Later, Löwe et al. (2022) extends the neural Granger causality idea to causal discovery on multiple samples with different causal relation-

ships but similar dynamics. However, all these approaches assume fully observed time-series and show inferior results given irregular data, which is shown in the experiments section. In this work, we leverage this Neural Granger Causal Discovery idea and build a two-stage iterative scheme to impute the unobserved data points and discover causal graphs jointly.

**Causal Discovery from Irregular Time-series.** Irregular time-series are very common in real scenarios, causal discovery addressing such data remains somewhat under-explored. When confronted with data missing, directly conducting causal inference might suffer from significant error (Runge, 2018a; Hyttinen et al., 2016). Although joint data imputation and causal discovery has been explored in static settings (Tu et al., 2019; Gain & Shpitser, 2018; Morales-Alvarez et al., 2022; Geffner et al., 2022), it is still under explored in time series causal discovery. There are mainly two solutions—either discovering causal relations with available observed incomplete data (Gain & Shpitser, 2018; Strobl et al., 2018) or filling missing values before causal discovery (Wang et al., 2020; Huang et al., 2020). To infer causal graphs from partially observed time-series, several algorithms are proposed, such as Expectation-Maximization approach (Gong et al., 2015), Latent Convergent Cross Mapping (Brouwer et al., 2021), Neural-ODE based approach (Bellot et al., 2022), Partial Canonical Correlation Analysis (Partial CCA), Generalized Lasso Granger (GLG) (Iseki et al., 2019), etc. Some other researchers introduce data imputation before causal discovery and have made progress recently. For example, Cao et al. (2018) learn to impute values via iteratively applying RNN and Cini et al. (2022) use Graph Neural Networks, while a recently proposed data completion method by Chen et al. (2022) uses Gaussian Process Regression. In this paper, we use a deep neural network similar to Cao et al. (2018)'s work, but differently, we propose to impute missing data points and discover causal graphs jointly instead of sequentially. Moreover, these two processes mutually improve each other and achieve high performance.

## 3 PROBLEM FORMULATION

### 3.1 NONLINEAR STRUCTURAL CAUSAL MODELS WITH IRREGULAR OBSERVATION

Let us denote by $\mathcal{X} = \{\boldsymbol{x}_{1:L,i}\}_{i=1}^{N}$ a uniformly sampled observation of a dynamic system, in which $\boldsymbol{x}_t$ represents the sample vector at time point $t$ and consists of $N$ variables $\{x_{t,i}\}$, with $t \in \{1, ..., L\}$ and $i \in \{1, ..., N\}$. In this paper, we adopt the representation proposed by Tank et al. (2022) and Khanna & Tan (2020), and assume each sampled variable $x_{t,i}$ be generated by the following model

$$x_{t,i} = f_i(\boldsymbol{x}_{t-\tau:t-1,1}, \boldsymbol{x}_{t-\tau:t-1,2}, ..., \boldsymbol{x}_{t-\tau:t-1,N}) + e_{t,i}, \quad i = 1, 2, ..., N. \tag{1}$$

Here $\tau$ denotes the maximal time lag. In this paper, we focus on dealing with causal inference from irregular time series, and use a bi-value observation mask $o_{t,i}$ to label the missing entries, i.e., the observed vector equals to its latent version when $o_{t,i}$ equals to 1: $\widetilde{x}_{t,i} \overset{\Delta}{=} x_{t,i} \cdot o_{t,i}$. In this paper we consider two types of recurrent data missing in practical observations:

**Random Missing.** The $i$th data point in the observations are missing with a certain probability $p_i$, here in our experiments the missing probability follows Bernoulli distribution $o_{t,i} \sim Ber(1 - p_i)$.

**Periodic Missing.** Different variables are sampled with their own periods $T_i$. We model the sampling process for $i$th variable with an observation function $o_{t,i} = \sum_{n=0}^{\infty} \delta(t - nT_i)$, $T_i = 1, 2, ...$ with $\delta(\cdot)$ denoting the Dirac's delta function.

### 3.2 NONLINEAR GRANGER CAUSALITY

For a dynamic system, time-series $i$ Granger causes time-series $j$ when the past values of time-series $\boldsymbol{x}_i$ aid in the prediction of the current and future status of time-series $\boldsymbol{x}_j$. The standard Granger causality is defined for linear relation scenarios, but recently extended to nonlinear relations:

**Definition 1** *Time-series i Granger cause j if and only if there exists $\boldsymbol{x}'_{t-\tau:t-1,i} \neq \boldsymbol{x}_{t-\tau:t-1,i}$,*

$$\begin{aligned} f_j(\boldsymbol{x}_{t-\tau:t-1,1}, ..., \boldsymbol{x}'_{t-\tau:t-1,i}, ..., \boldsymbol{x}_{t-\tau:t-1,N}) \neq \\ f_j(\boldsymbol{x}_{t-\tau:t-1,1}, ..., \boldsymbol{x}_{t-\tau:t-1,i}, ..., \boldsymbol{x}_{t-\tau:t-1,N}) \end{aligned} \tag{2}$$

*i.e., the past data points of time-series i influence the prediction of $x_{t,j}$.*

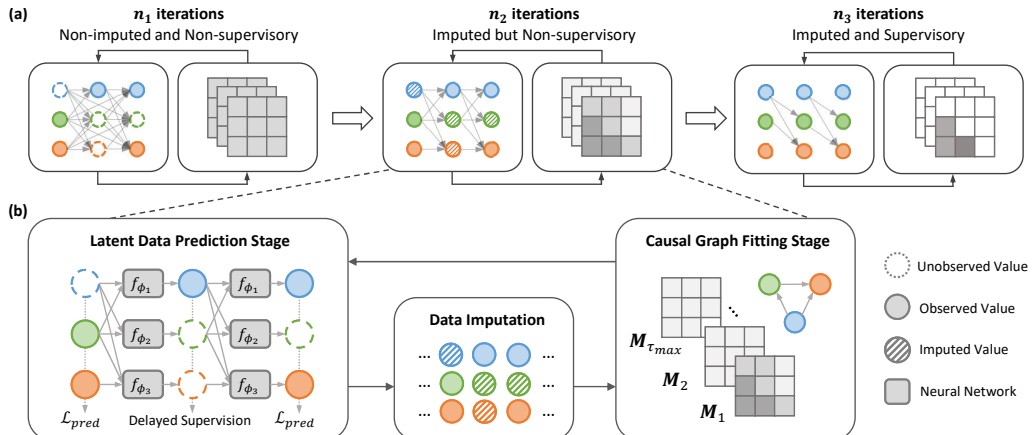

Figure 1: Illustration of the proposed CUTS, with a 3-variable example. **(a)** Illustration of our learning strategy described in Section 4.3, with three groups of iterations being of the same alternation scheme shown in (b) but different settings in data imputation and supervised model learning. **(b)** Illustration of each iteration in CUTS. The dynamics reflected by the observed time-series $x_1$ and $x_2$ are described by DSGNN in the *Latent data prediction stage* (left). With the modeled dynamics, unobserved data points are imputed (center) and fed into the *Causal graph fitting stage* for an improved graph inference (right).

Granger causality is highly compatible with neural networks (NN). Considering the universal approximation ability of NN (Hornik et al., 1989), it is possible to fit a causal relationship function with component-wise MLPs or RNNs. Imposing a sparsity regularizer onto the weights of network connections, as mentioned by Tank et al. (2022) and Khanna & Tan (2020), NNs can learn the causal relationships among all $N$ variables. The inferred pair-wise Granger causal relationships can then be aggregated into a Directed Acyclic Graph (DAG), represented as an adjacency matrix $\boldsymbol{A} = \{a_{ij}\}_{i,j=1}^{N}$, where $a_{ij} = 1$ denotes time-series $i$ Granger causes $j$ and $a_{ij} = 0$ means otherwise. This paradigm is well explored and shows convincing empirical evidence in recent years (Tank et al., 2022; Khanna & Tan, 2020; Löwe et al., 2022).

Although Granger causality is not necessarily the true causality, Peters et al. (2017) provide justification of (time-invariant) Granger causality when assuming no unobserved variables and no instantaneous effects, as is mentioned by Löwe et al. (2022) and Vowels et al. (2021).

In this paper, we propose a new inference approach to successfully identify causal relationships from irregular time-series data.

# 4 IRREGULAR TIME-SERIES CAUSAL DISCOVERY

CUTS implements the causal graph as a set of Causal Probability Graphs (CPGs) $\mathcal{G} = \langle \mathcal{X}, \{\boldsymbol{M}_\tau\}_{\tau=0}^{\tau_{max}} \rangle$ where the element $m_{\tau,ij} \in \boldsymbol{M}_\tau$ represents the probability of causal influence from $x_{t-\tau,i}$ to $x_{t,j}$, i.e. $m_{\tau,ij} = p(x_{t-\tau,i} \rightarrow x_{t,j})$. Since we assume no instantaneous effects, time-series $i$ Granger cause $j$ if and only if there exist causal relations on at least one time lag, we define our discovered causal graph $\tilde{\boldsymbol{A}}$ to be the maximum value across all time lags $\tau \in \{1, ..., \tau_{max}\}$

$$\tilde{a}_{i,j} = \max\left(m_{1,ij}, ..., m_{\tau_{max},ij}\right). \tag{3}$$

Specifically, if $\tilde{a}_{i,j}$ is penalized to zero (or below certain threshold), we deduce that time-series $i$ does not influence the prediction of time-series $j$, i.e., $i$ does not Granger cause $j$.

During training, we alternatively learn the prediction model and CPG matrix, which are respectively implemented by *Latent data prediction stage* and *Causal graph fitting stage*. Besides, proper learning strategies are designed to facilitate convergence.

## 4.1 LATENT DATA PREDICTION STAGE

The proposed *Latent data prediction stage* is designed to fit the data generation function for time-series $i$ with a neural network $f_{\phi_i}$, which takes into account its parent nodes in the causal graph. Here we propose Delayed Supervision Graph Neural Network (DSGNN) for imputing the missing entries in the observation.

The inputs to DSGNN include all the historical data points (with a maximum time lag $\tau_{max}$) $\boldsymbol{x}_{t-\tau:t-1,i}$ and the discovered CPGs. During training we sample the causal graph with Bernoulli distribution, in a similar manner to Lippe et al. (2021)'s work, and the prediction $\hat{\boldsymbol{x}}$ is the output of the neural network $f_{\phi_i}$

$$\hat{x}_{t,i} = f_{\phi_i}(\mathcal{X} \odot \boldsymbol{S}) = f_{\phi_i}(\boldsymbol{x}_{t-\tau:t-1,1} \odot \boldsymbol{s}_{1:\tau,1i}, ..., \boldsymbol{x}_{t-\tau:t-1,N} \odot \boldsymbol{s}_{1:\tau,Ni}), \quad (4)$$

where $\boldsymbol{S} = \{\boldsymbol{S}_\tau\}_{\tau=1}^{\tau=\tau_{max}}$, and $\boldsymbol{s}_{\tau,ij} \sim Ber(1 - m_{\tau,ij})$ and $\odot$ denotes the Hadamard product. $\boldsymbol{S}$ is sampled for each training sample in a mini-batch. The fitting is done under supervision from the observed data points. Specifically, we update the network parameters $\phi_i$ by minimizing the following loss function

$$\mathcal{L}_{pred}\left(\tilde{\mathcal{X}}, \hat{\mathcal{X}}, \mathcal{O}\right) = \sum_{i=1}^{N} \frac{\langle \mathcal{L}_2\left(\hat{\boldsymbol{x}}_{1:L,i}, \tilde{\boldsymbol{x}}_{1:L,i}\right), \boldsymbol{o}_{1:L,i}\rangle}{\frac{1}{L}\langle \boldsymbol{o}_{1:L,i}, \boldsymbol{o}_{1:L,i}\rangle} \quad (5)$$

where $\boldsymbol{o}_i$ denotes the observation mask, $\langle \cdot \rangle$ is the dot product, and $\mathcal{L}_2$ represents the MSE loss function. Then, the data imputation is performed with the following equation

$$\tilde{x}_{t,i}^{(m+1)} = \begin{cases} (1-\alpha)\tilde{x}_{t,i}^{(m)} + \alpha\hat{x}_{t,i}^{(m)} & o_{t,i} = 0 \text{ and } m \geq n_1 \\ \tilde{x}_{t,i}^0 & o_{t,i} = 1 \text{ or } m < n_1 \end{cases} \quad (6)$$

Here $m$ indexes the iteration steps, and $\tilde{x}_{t,i}^{(0)}$ denotes the initial data (unobserved entries filled with zero order holder). $\alpha$ is selected to prevent the abrupt change of imputed data. For the missing points, their predicted value $\hat{x}_{t,i}^{(m)}$ is unsupervised with $\mathcal{L}$ but updated to $\tilde{x}_{t,i}^{(m)}$ to obtain a "delayed" error in causal graph inference. Moreover, we impute the missing values with the help of discovered CPG $\mathcal{G}$ (sampled with Bernoulli Distribution), as illustrated in Figure 1 (b), which is proved to significantly improve performance in experiments.

## 4.2 CAUSAL GRAPH DISCOVERY STAGE

After imputing the missing time-series, we proceed to learn CPG in the *Causal graph fitting stage*, to determine the causal probability $p(x_{t-\tau,i} \rightarrow x_{t,j}) = m_{\tau,ij}$, we model this likelihood with $m_{\tau,ij} = \sigma(\theta_{\tau,ij})$ where $\sigma(\cdot)$ denotes the sigmoid function and $\boldsymbol{\theta}$ is the learned parameter set. Since we assume no instantaneous effect, it is unnecessary to learn the edge direction in CPG.

In this stage we optimize the graph parameters $\boldsymbol{\theta}$ by minimizing the following objective

$$\mathcal{L}_{graph}\left(\tilde{\mathcal{X}}, \hat{\mathcal{X}}, \mathcal{O}, \boldsymbol{\theta}\right) = \mathcal{L}_{pred}\left(\tilde{\mathcal{X}}, \hat{\mathcal{X}}, \mathcal{O}\right) + \lambda||\sigma(\boldsymbol{\theta})||_1, \quad (7)$$

where $\mathcal{L}_{pred}$ is the squared error loss penalizing prediction error defined in Equation (5) and $|| \cdot ||_1$ being the $\mathcal{L}_1$ regularizer to enforce sparse connections on the learned CPG. If $\forall \tau \in [1, \tau_{max}], \theta_{\tau,ij}$ are penalized to $-\infty$ (and $m_{\tau,ij} \rightarrow 0$), then we deduce that time-series $i$ does not Granger cause $j$.

## 4.3 THE LEARNING STRATEGY.

The overall learning process consists of $n = n_1 + n_2 + n_3$ epochs, which is illustrated in Figure 1 (a): in the first $n_1$ epochs DSGNN and CPG are optimized without data imputation (missing entries are set with initial guess); in the next $n_2$ epochs the iterative model learning continues with data imputation, but the imputed data are not used for model supervision; for the last $n_3$ epochs the learned CPG is refined based on supervision from all the data points (including the imputed ones).

**Fine-tuning.** The main training process is the alternation between *Latent data prediction stage* and *Causal graph fitting stage*. Considering that after sufficient iterations (here $n_1 + n_2$) the unobserved data points can be reliably imputed with the discovery of causal relations, and we can incorporate these predicted points to supervise the model and fine-tune the parameters to improve the performance further. In the last $n_3$ epochs CPG is optimized with the loss function

$$\mathcal{L}_{ft}\left(\tilde{\mathcal{X}}, \hat{\mathcal{X}}\right) = \mathcal{L}_2\left(\hat{\boldsymbol{x}}_{1:L,i}, \tilde{\boldsymbol{x}}_{1:L,i}\right) + \lambda||\sigma(\boldsymbol{\theta})||_1. \quad (8)$$

**Parameter Settings.** During training the $\tau$ value for Gumbel Softmax is initially set to a relatively high value and annealed to a low value in the first $n_1 + n_2$ epochs and then reset for the last $n_3$ epochs. The learning rates for *Latent data prediction stage* and *Causal graph fitting stage* are respectively set as $lr_{data}$ and $lr_{graph}$ and gradually scheduled to $0.1lr_{data}$ and $0.1lr_{graph}$ during all $n_1 + n_2 + n_3$ epochs. The detailed hyperparameter settings are listed in Appendix Section A.3.

### 4.4 CONVERGENCE CONDITIONS FOR GRANGER CAUSALITY.

We show in Theorem 1 that under certain assumptions, the discovered causal adjacency matrix will converge to the true Granger causal matrix.

**Theorem 1** *Given a time-series dataset $\mathcal{X} = \{\boldsymbol{x}_{1:L,i}\}_{i=1}^{N}$ generated with Equation 1, we have*

1. *$\exists \lambda, \forall \tau \in \{1,.., \tau_{max}\}$, causal probability matrix element $m_{\tau,ij} = \sigma(\theta_{\tau,ij})$ converges to 0 if time-series $i$ does not Granger cause $j$, and*

2. *$\exists \tau \in \{1,.., \tau_{max}\}, m_{\tau,ij}$ converges to 1 if time-series $i$ Granger cause $j$,*

*if the following two conditions hold:*

1. *DSGNN $f_{\phi_i}$ in Latent data prediction stage model generative function $f_i$ with an error smaller than arbitrarily small value $e_{NN,i}$;*

2. *$\exists \lambda_0, \forall i, j = 1, ..., N, \|f_{\phi_j}(\mathcal{X} \odot \boldsymbol{S}_{\tau,ij=1}) - f_{\phi_j}(\mathcal{X} \odot \boldsymbol{S}_{\tau,ij=0})\|_2^2 > \lambda_0$, where $\boldsymbol{S}_{\tau,ij=l}$ is set $\boldsymbol{S}$ with element $\boldsymbol{s}_{\tau,ij} = l$.*

The implications behind these two conditions can be intuitively explained. Assumption 1 is intrinsically the Universal Approximation Theorem (Hornik et al., 1989) of neural network, i.e., the network is of an appropriate structure and fed with sufficient training data. Assumption 2 means there exists a threshold $\lambda_0$ to binarize $\|f_{\phi_i}(X \odot S_{\tau,ij=1}) - f_{\phi_i}(X \odot S_{\tau,ij=0})\|$, serving as an indicator as to whether time-series $j$ contributes to prediction of $i$.

The proof of Theorem 1 is detailed in Appendix Section A.1. Although the convergence condition is relevant to the appropriate setting of $\lambda$, we will show in Appendix Section A.4.6 that our algorithm is robust to the setting changes of $\lambda$ over a wide range.

## 5 EXPERIMENTS

**Datasets.** We evaluate the performance of the proposed causal discovery approach CUTS on both numerical simulation and real-scenario inspired data. The simulated datasets come from a linear Vector Autoregressive (VAR) model and a nonlinear Lorenz-96 model (Karimi & Paul, 2010), while the real-scenario inspired datasets are from NetSim (Smith et al., 2011), an fMRI dataset describing the connecting dynamics of 15 human brain regions. The irregular observations are generated according to the following mechanisms: Random Missing (RM) is simulated by sampling over a uniform distribution with missing probability $p_i$; Periodic Missing (PM) is simulated with sampling period $T_i$ randomly chosen for each time-series with the maximum period being $T_{max}$. For statistical quantitative evaluation of different causal discovery algorithms, we take average over multiple $p_i$ and $T_i$ in our experiments.

**Baseline Algorithms.** To demonstrate the superiority of our approach, we compare with five baseline algorithms: (i) Neural Granger Causality (NGC, Tank et al. (2022)), which utilizes MLPs and RNNs combined with weight penalties to infer Granger causal relationships, in the experiments we use the component-wise MLP model; (ii) economy-SRU (eSRU, Khanna & Tan (2020)), a variant of SRU that is less prone to over-fitting when inferring Granger causality; (iii) PCMCI (proposed by Runge et al.), a non-Granger-causality-based method in which we use conditional independence tests provided along with its repository[1], i.e., ParCorr (linear partial correlation) for conditional independence tests for linear scenarios and GPDC (Gaussian Process regression and Distance Correlation Rasmussen (2003) Székely et al. (2007)) for nonlinear scenarios. (iv) Latent Convergent Cross Mapping (LCCM, Brouwer et al. (2021)), a CCM-based approach that also tackles the irregular time-series problem. (v) Neural Graphical Model (NGM, Bellot et al. (2022)) which is based on Neural Ordinary Differential Equations (Neural-ODE) to solve the irregular time-series problem. In terms of quantitative evaluation, we use area under the ROC curve (AUROC) as the criterion. For NGC, AUROC values are computed by running the algorithm with $\lambda$ varying within a range of values. For eSRU, PCMCI, LCCM, and NGM, the AUROC values are obtained with different thresholds. For a fair comparison, we applied parameter searching to determine the hyperparameters

---

[1]https://github.com/jakobrunge/tigramite

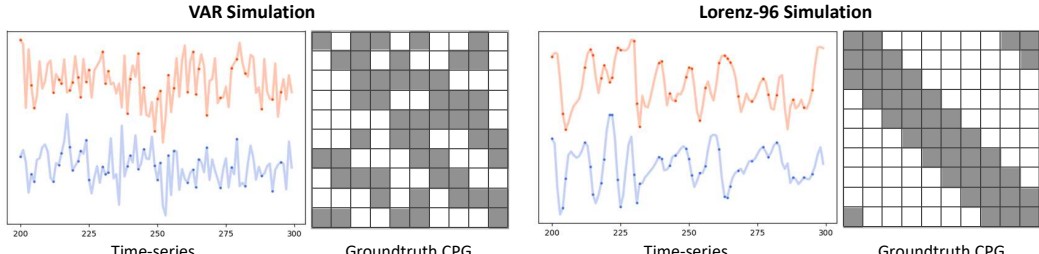

Figure 2: Examples of our simulated VAR and Lorenz-96 datasets, with two of the total 10 generated time-series from the groundtruth CPG plotted as orange and blue solid lines, while the non-uniformly sampled points are labeled with scattered points.

of the baseline algorithms with the best performance. For baseline algorithms unable to handle irregular time-series data, i.e., NGC, PCMCI, and eSRU, we imputed the irregular time-series before feeding them to causal discovery modules, and use three data imputation algorithms, i.e., Zero-order Holder (ZOH), Gaussian Process Regression (GP), and Multivariate Time Series Imputation by Graph Neural Network (GRIN, Cini et al. (2022)).

## 5.1 VAR SIMULATION DATASETS

VAR datasets are simulated following

$$\boldsymbol{x}_t = \sum_{\tau=1}^{\tau_{max}} \boldsymbol{A}_\tau \boldsymbol{x}_{t-\tau} + e_t, \tag{9}$$

where the matrix $\boldsymbol{A}_\tau$ is the sparse autoregressive coefficients for time lag $\tau$. Time-series $i$ Granger cause time-series $j$ if $\exists \tau \in \{1, ..., \tau_{max}\}, a_{\tau,ij} > 0$. The objective of causal discovery is to reconstruct the non-zero elements in causal graph $\boldsymbol{A}$ (where each element of $\boldsymbol{A}$ $a_{ij} = \max(a_{1,ij}, ..., a_{\tau_{max},ij})$) with $\tilde{\boldsymbol{A}}$. We set $\tau_{max} = 3$, $N = 10$ and time-series length $L = 10000$ in this experiment. For missing mechanisms, we set $p = 0.3, 0.6$, respectively for Random Missing and $T_{max} = 2, 4$ respectively for Periodic Missing. Experimental results are shown in the upper half of Table 1. We can see that CUTS beats PCMCI, NGC, and eSRU combined with ZOH, GP, and GRIN in most cases, except for the case of VAR with random missing ($p = 0.3$) where PCMCI + GRIN is better by only a small margin (+0.0012). The superiority is especially prominent when with a larger percentage of missing values ($p = 0.6$ for random missing and $T_{max} = 4$ for periodic missing). Differently, data imputation algorithms GP and GRIN provide performance gain in some scenarios but fail to boost causal discovery in others. This indicates that simply combining previous data imputation algorithms with causal discovery algorithms cannot give stable and promising results, and is thus less practical than our approach. We also beat LCCM and NGM which originally tackles the irregular time series problem by a clear margin. This hampered performance may be attributed to the fact that LCCM and NGM both utilize Neural-ODE to model the dynamics and do not cope with VAR datasets well.

## 5.2 LORENZ-96 SIMULATION DATASETS

Lorenz-96 datasets are simulated according to

$$\frac{dx_{t,i}}{dt} = -x_{t,i-1}(x_{t,i-2} - x_{t,i+1}) - x_{t,i} + F, \tag{10}$$

where $-x_{t,i-1}(x_{t,i-2} - x_{t,i+1})$ is the advection term, $x_{t,i}$ is the diffusion term, and $F$ is the external forcing (a larger $F$ implies a more chaotic system). In this Lorenz-96 model each time-series $\boldsymbol{x}_i$ is affected by historical values of four time-series $\boldsymbol{x}_{i-2}, \boldsymbol{x}_{i-1}, \boldsymbol{x}_i, \boldsymbol{x}_{i+1}$, and each row in the ground truth causal graph $\boldsymbol{A}$ has four non-zero elements. Here we set the maximal time-series length $L = 1000, N = 10$, force constant $F = 10$ and show experimental results for $F = 40$ in the Appendix Section A.4.7. From the results in the lower half of Table 1, one can draw similar conclusions to those on VAR datasets: CUTS outperforms baseline causal discovery methods either with or without data imputation.

## 5.3 NETSIM DATASETS

Table 1: Performance comparison of CUTS with (i) PCMCI, eSRU, NGC combined with imputation method ZOH, GP, GRIN and (ii) LCCM, NGM which do not need data imputation. Experiments are performed on VAR and Lorenz-96 datasets in terms of AUROC. Results are averaged over 10 randomly generated datasets.

| Methods | Imputation | VAR with Random Missing | | VAR with Periodic Missing | |
|---|---|---|---|---|---|
| | | $p = 0.3$ | $p = 0.6$ | $T_{max} = 2$ | $T_{max} = 4$ |
| PCMCI | ZOH | $0.9904 \pm 0.0078$ | $0.9145 \pm 0.0204$ | $0.9974 \pm 0.0040$ | $0.9787 \pm 0.0196$ |
| | GP | $0.9930 \pm 0.0072$ | $0.8375 \pm 0.0651$ | $0.9977 \pm 0.0038$ | $0.9332 \pm 0.1071$ |
| | GRIN | $\mathbf{0.9983 \pm 0.0028}$ | $0.9497 \pm 0.0132$ | $\mathbf{0.9989 \pm 0.0017}$ | $0.9774 \pm 0.0169$ |
| NGC | ZOH | $0.9899 \pm 0.0105$ | $0.9325 \pm 0.0266$ | $0.9808 \pm 0.0117$ | $0.9439 \pm 0.0264$ |
| | GP | $0.9821 \pm 0.0097$ | $0.5392 \pm 0.1176$ | $0.9833 \pm 0.0108$ | $0.7350 \pm 0.2260$ |
| | GRIN | $0.8186 \pm 0.1720$ | $0.5918 \pm 0.1170$ | $0.8621 \pm 0.0661$ | $0.6677 \pm 0.1350$ |
| eSRU | ZOH | $0.9760 \pm 0.0113$ | $0.8464 \pm 0.0299$ | $0.9580 \pm 0.0276$ | $0.9214 \pm 0.0257$ |
| | GP | $0.9747 \pm 0.0096$ | $0.8988 \pm 0.0301$ | $0.9587 \pm 0.0191$ | $0.8166 \pm 0.1085$ |
| | GRIN | $0.9677 \pm 0.0134$ | $0.8399 \pm 0.0242$ | $0.9740 \pm 0.0150$ | $0.8574 \pm 0.0869$ |
| LCCM | | $0.6851 \pm 0.0411$ | $0.6530 \pm 0.0212$ | $0.6462 \pm 0.0225$ | $0.6388 \pm 0.0170$ |
| NGM | | $0.7608 \pm 0.0910$ | $0.6350 \pm 0.0770$ | $0.8596 \pm 0.0353$ | $0.7968 \pm 0.0305$ |
| **CUTS (Proposed)** | | $\mathbf{0.9971 \pm 0.0026}$ | $\mathbf{0.9766 \pm 0.0074}$ | $\mathbf{0.9992 \pm 0.0016}$ | $\mathbf{0.9958 \pm 0.0069}$ |
| Methods | Imputation | Lorenz-96 with Random Missing | | Lorenz-96 with Periodic Missing | |
| | | $p = 0.3$ | $p = 0.6$ | $T_{max} = 2$ | $T_{max} = 4$ |
| PCMCI | ZOH | $0.8173 \pm 0.0491$ | $0.7275 \pm 0.0534$ | $0.7229 \pm 0.0348$ | $0.7178 \pm 0.0668$ |
| | GP | $0.7545 \pm 0.0585$ | $0.7862 \pm 0.0379$ | $0.7782 \pm 0.0406$ | $0.7676 \pm 0.0360$ |
| | GRIN | $0.8695 \pm 0.0301$ | $0.7544 \pm 0.0404$ | $0.7299 \pm 0.0545$ | $0.7277 \pm 0.0947$ |
| NGC | ZOH | $0.9933 \pm 0.0058$ | $0.9526 \pm 0.0220$ | $0.9903 \pm 0.0096$ | $0.9776 \pm 0.0120$ |
| | GP | $0.9941 \pm 0.0064$ | $0.5000 \pm 0.0000$ | $0.9949 \pm 0.0050$ | $0.7774 \pm 0.2300$ |
| | GRIN | $0.9812 \pm 0.0105$ | $0.7222 \pm 0.0680$ | $0.9640 \pm 0.0193$ | $0.8430 \pm 0.0588$ |
| eSRU | ZOH | $0.9968 \pm 0.0038$ | $0.9089 \pm 0.0261$ | $0.9958 \pm 0.0031$ | $0.9815 \pm 0.0148$ |
| | GP | $0.9977 \pm 0.0035$ | $0.9597 \pm 0.0169$ | $0.9990 \pm 0.0015$ | $0.9628 \pm 0.0371$ |
| | GRIN | $0.9937 \pm 0.0071$ | $0.9196 \pm 0.0251$ | $0.9873 \pm 0.0110$ | $0.8400 \pm 0.1451$ |
| LCCM | | $0.7168 \pm 0.0245$ | $0.6685 \pm 0.0311$ | $0.7064 \pm 0.0324$ | $0.7129 \pm 0.0235$ |
| NGM | | $0.9180 \pm 0.0199$ | $0.7712 \pm 0.0456$ | $0.9751 \pm 0.0112$ | $0.9171 \pm 0.0189$ |
| **CUTS (Proposed)** | | $\mathbf{0.9996 \pm 0.0005}$ | $\mathbf{0.9705 \pm 0.0118}$ | $\mathbf{1.0000 \pm 0.0000}$ | $\mathbf{0.9959 \pm 0.0042}$ |

To validate the performance of CUTS on real-scenario data, We use data from 10 humans in NetSim datasets[2], which is generated with synthesized dynamics of brain region connectivity and unknown to us and the algorithm. The total length of each time-series data $L = 200$ and the number of time-series $N = 15$. By testing our CUTS on this dataset we show that our algorithm is capable of discovering causal relations with irregular time-series data for scientific discovery. However, $L = 200$ is a small data size, therefore we only perform experiments with the Random Missing situation. Experimental results shown in Table 2 tell that our approach beats all existing methods on both missing proportions.

Table 2: Quantitative results on NetSim dataset. Results averaged over 10 human brain subjects.

| Met. | Imp. | NetSim with Random Missing | |
|---|---|---|---|
| | | $p = 0.1$ | $p = 0.2$ |
| PCMCI | ZOH | $0.7625 \pm 0.0539$ | $0.7455 \pm 0.0675$ |
| | GP | $0.7462 \pm 0.0396$ | $0.7551 \pm 0.0451$ |
| | GRIN | $0.7475 \pm 0.0517$ | $0.7353 \pm 0.0611$ |
| NGC | ZOH | $0.7656 \pm 0.0576$ | $\mathbf{0.7668 \pm 0.0403}$ |
| | GP | $0.7506 \pm 0.0532$ | $0.7545 \pm 0.0518$ |
| | GRIN | $0.6744 \pm 0.0743$ | $0.5826 \pm 0.0476$ |
| eSRU | ZOH | $0.6384 \pm 0.0473$ | $0.6592 \pm 0.0248$ |
| | GP | $0.6147 \pm 0.0454$ | $0.6330 \pm 0.0449$ |
| | GRIN | $0.6141 \pm 0.0529$ | $0.5818 \pm 0.0588$ |
| LCCM | | $0.7711 \pm 0.0301$ | $0.7594 \pm 0.0246$ |
| NGM | | $0.7417 \pm 0.0380$ | $0.7215 \pm 0.0330$ |
| **CUTS** | | $\mathbf{0.7948 \pm 0.0381}$ | $\mathbf{0.7699 \pm 0.0550}$ |

## 5.4 ABLATION STUDIES

Besides demonstrating the advantageous performance of the final results, we further conducted a series of ablation studies to quantitatively evaluate the contributions of the key technical designs or learning strategies in CUTS. Due to page limit, we only show experiments on Lorenz-96 datasets with Random Missing settings in this section, and leave the other results in the Appendix Section A.4.2.

**Causal Discovery Boosts Data Imputation.** To validate that *Latent data prediction stage* helps *Causal graph fitting stage*, we reset CPGs $M_\tau^m$ to all-one matrices in *Latent data prediction*

---

[2]Shared at https://www.fmrib.ox.ac.uk/datasets/netsim/sims.tar.gz

Table 3: Quantitative results of ablation studies. "CUTS (Full)" denotes the default settings in this paper. Here we run experiments on Lorenz-96 datasets. Ablation study results on other datasets are provided in Appendix Section A.4.2.

| Methods | Lorenz-96 with Random Missing | | Lorenz-96 with Periodic Missing | |
|---|---|---|---|---|
| | $p = 0.3$ | $p = 0.6$ | $T_{max} = 2$ | $T_{max} = 4$ |
| **CUTS (Full)** | **$0.9996 \pm 0.0005$** | **$0.9705 \pm 0.0118$** | **$1.0000 \pm 0.0000$** | **$0.9959 \pm 0.0042$** |
| ZOH for Imputation | $0.9799 \pm 0.0071$ | $0.8731 \pm 0.0312$ | $0.9981 \pm 0.0021$ | $0.9865 \pm 0.0128$ |
| GP for Imputation | $0.9863 \pm 0.0058$ | $0.8575 \pm 0.0536$ | $0.9965 \pm 0.0036$ | $0.9550 \pm 0.0407$ |
| GRIN for Imputation | $0.9793 \pm 0.0126$ | $0.8983 \pm 0.0299$ | $0.9869 \pm 0.0101$ | $0.9325 \pm 0.0415$ |
| No Imputation | $0.9898 \pm 0.0045$ | $0.9206 \pm 0.0216$ | $0.9968 \pm 0.0032$ | $0.9797 \pm 0.0204$ |
| Remove CPG for Imput. | $0.9972 \pm 0.0021$ | $0.9535 \pm 0.0167$ | $0.9989 \pm 0.0011$ | $0.9926 \pm 0.0045$ |
| No Finetuning Stage | $0.9957 \pm 0.0036$ | $0.9665 \pm 0.0096$ | $0.9980 \pm 0.0025$ | $0.9794 \pm 0.0124$ |

*stage* and then $\hat{x}_{t,i}$ is predicted with all time-series instead of only the parent nodes. This experiment is shown as "Remove CPG for Imput." in Table 6. It is observed that introducing CPGs in data imputation is especially helpful with large quantities of missing values ($p = 0.6$ for Random Missing or $T_{max} = 4$ for Periodic Missing). Comparing with the scores in the first row, we can see that introducing CPGs in data imputation boosts AUROC by $0.0011 \sim 0.0170$.

**Data Imputation Boosts Causal Discovery.** To show that *Causal graph fitting stage* helps *Latent data prediction stage*, we disable data imputation operation defined in Equation 6, i.e., $\alpha = 0$. In other words, *Causal graph fitting stage* is performed with just the initially filled data (Appendix Section A.3.2), with the results shown as "No Imputation" in Table 6. Compared with the first row, we can see that introducing data imputation boosts AUROC by $0.0032 \sim 0.0499$. We further replace our data imputation module with baseline modules (ZOH, GP, GRIN) to show the effectiveness of our design. It is observed that our algorithm beats "ZOH for Imputation", "GP for Imputation", "GRIN for Imputation" in most scenarios.

**Finetuning Stage Raises Performance.** We disable the finetuning stage and find that the performance drops slightly, as shown in the "No Finetuning Stage" row in Table 6. In other words, the finetuning stage indeed helps to refine the causal discovery process.

## 5.5 Additional Experiments

We further conduct additional experiments in Appendix to show experiments on more datasets (Appendix Section A.4.1), ablation study for choice of epoch numbers (Appendix Section A.4.3), ablation study results on VAR and NetSim datasets (Appendix Section A.4.2), performance on 3-dimensional temporal causal graph (Appendix Section A.4.4), CUTS's performance superiority on regular time-series (Appendix Section A.4.5), robustness to different noise levels (Appendix Section A.4.8), robustness to hyperparameter settings (Appendix Section A.4.6), and results on Lorenz-96 with forcing constant $F = 40$ (Appendix Section A.4.7). We further provide implementation details and hyperparameters settings of CUTS and baseline algorithms in Appendix Section A.3, and the pseudocode of our approach in Appendix Section A.5.

## 6 Conclusions

In this paper we propose CUTS, a time-series causal discovery method applicable for scenarios with irregular observations with the help of nonlinear Granger causality. We conducted a series of experiments on multiple datasets with Random Missing as well as Periodic Missing. Compared with previous methods, CUTS utilizes two alternating stages to discover causal relations and achieved superior performance. We show in the ablation section that these two stages mutually boost each other to achieve an improved performance. Moreover, our CUTS is widely applicable for time-series with different lengths, scales well to large sets of variables, and is robust to noise. Our code is publicly available at `https://github.com/jarrycyx/unn`.

In this work we assume no latent confounder and no instantaneous effect for Granger causality. Our future works includes: (i) Causal discovery in the presence of latent confounder or instantaneous effect. (ii) Time-series imputation with causal models.

## REPRODUCIBILITY STATEMENT

For the purpose of reproducibility, we include the source code in the supplementary files, and will published on GitHub upon acceptance. Datasets generation process is also included in source code. Moreover, we provide all hyperparameters used for all methods in Appendix Section A.4.6. The experiments are deployed on a server with Intel Core CPU and NVIDIA RTX3090 GPU.

## ACKNOWLEDGMENTS

This work is jointly funded by Ministry of Science and Technology of China (Grant No. 2020AAA0108202), National Natural Science Foundation of China (Grant No. 61931012 and 62088102), Beijing Natural Science Foundation (Grant No. Z200021), and Project of Medical Engineering Laboratory of Chinese PLA General Hospital (Grant No. 2022SYSZZKY21).

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

# A APPENDIX

## CONTENTS

## A.1 CONVERGENCE CONDITIONS FOR GRANGER CAUSALITY

### A.1.1 PROOF OF THEOREM 1

We proved that in Theorem 1 our CUTS can discover the correct Granger causality with the following assumptions:

1. DSGNN $f_{\phi_i}$ in *Latent data prediction stage* model generative function $f_i$ with an error smaller than arbitrarily small value $e_{\text{NN},i}$;

2. $\exists \lambda_0, \forall i, j = 1, ..., N, \|f_{\phi_j}(\mathcal{X} \odot \boldsymbol{S}_{\tau,ij=1}) - f_{\phi_j}(\mathcal{X} \odot \boldsymbol{S}_{\tau,ij=0})\|_2^2 > \lambda_0$, where $\boldsymbol{S}_{\tau,ij=l}$ is set $\boldsymbol{S}$ with element $\boldsymbol{s}_{\tau,ij} = l$.

In *Causal graph fitting stage* the loss function

$$
\begin{aligned}
\mathcal{L}_{graph}(\tilde{\mathcal{X}}, \hat{\mathcal{X}}, \mathcal{O}, \boldsymbol{\theta}) &= \sum_{i=1}^{N} \frac{\langle \mathcal{L}_2(\hat{\boldsymbol{x}}_{1:L,i}, \tilde{\boldsymbol{x}}_{1:L,i}), \boldsymbol{o}_i \rangle}{\frac{1}{L}\langle \boldsymbol{o}_{1:L,i}, \boldsymbol{o}_{1:L,i} \rangle} + \lambda ||\sigma(\boldsymbol{\theta})||_1 \\
&= \sum_{i=1}^{N} \sum_{t=1}^{L} c_i o_{t,i}(\boldsymbol{x}_{t,i} - f_{\phi_j}(\mathcal{X} \odot \boldsymbol{S}))^2 + \lambda ||\sigma(\boldsymbol{\theta})||_1
\end{aligned}
\tag{11}
$$

where $s_{\tau,ij} \sim Ber(\sigma(\theta_{\tau,ij}))$, $c_i = \frac{L}{\langle \boldsymbol{o}_{1:L,i}, \boldsymbol{o}_{1:L,i}\rangle}$. We use the REINFORCE (Williams, 1992) trick and $m_{\tau,ij}{}'s$ gradient is calculated as

$$
\begin{aligned}
\frac{\partial}{\partial\theta_{\tau,ij}}\mathbb{E}_{s_{\tau,ij}}[\mathcal{L}_{graph}] &= \mathbb{E}_{s_{\tau,ij}}[c_i o_{t,i}(\boldsymbol{x}_{t,j} - f_{\phi_j}(\mathcal{X}\odot\boldsymbol{S}))^2 \frac{\partial}{\partial\theta_{\tau,ij}}\log p_{s_{\tau,ij}}] + \lambda\sigma'(\theta_{\tau,ij}) \\
&= \lambda\sigma'(\theta_{\tau,ij}) + \sigma(\theta_{\tau,ij})c_i o_{t,i}(\boldsymbol{x}_{t,j} - f_{\phi_j}(\mathcal{X}\odot\boldsymbol{S}_{\tau,ij=1}))^2\frac{1}{\sigma(\theta_{\tau,ij})}\sigma'(\theta_{\tau,ij}) \\
&\quad + (1-\sigma(\theta_{\tau,ij}))c_i o_{t,i}(\boldsymbol{x}_{t,j} - f_{\phi_j}(\mathcal{X}\odot\boldsymbol{S}_{\tau,ij=0}))^2\frac{1}{\sigma(\theta_{\tau,ij})-1}\sigma'(\theta_{\tau,ij}) \\
&= \sigma'(\theta_{\tau,ij})(c_i o_{t,i}(\boldsymbol{x}_{t,j} - f_{\phi_j}(\mathcal{X}\odot\boldsymbol{S}_{\tau,ij=1}))^2 \\
&\quad - c_i o_{t,i}(\boldsymbol{x}_{t,j} - f_{\phi_j}(\mathcal{X}\odot\boldsymbol{S}_{\tau,ij=0}))^2 + \lambda).
\end{aligned}
\tag{12}
$$

Where $\boldsymbol{S}_{\tau,ij=l}$ denotes $\boldsymbol{S} = \{\boldsymbol{S}_\tau\}_{\tau=1}^{\tau_{max}}$ with $s_{\tau,ij}$ set to $l$, and $f_{\phi_j}(\mathcal{X}\odot\boldsymbol{S}_{\tau,ij=1}) = f_{\phi_j}(\boldsymbol{x}_{t-\tau:t-1,1}\odot \boldsymbol{s}_{1:\tau,1i},...,\boldsymbol{x}_{t-\tau:t-1,N}\odot\boldsymbol{s}_{1:\tau,Ni})$. According to Definition 1, time-series $i$ does not Granger cause $j$ if $\forall\tau \in \{1,...,\tau_{max}\}$, $\boldsymbol{x}_{t-\tau,i}$ is invariant of the prediction of $\boldsymbol{x}_{t,j}$. Then we have $\forall\tau \in \{1,...,\tau_{max}\}$, $f_{\phi_j}(...,\boldsymbol{x}_{t-\tau,i},...) = f_{\phi_j}(...,0,...)$, i.e., $f_{\phi_j}(\mathcal{X}\odot\boldsymbol{S}_{\tau,ij=1}) = f_{\phi_j}(\mathcal{X}\odot\boldsymbol{S}_{\tau,ij=0})$.

Applying additive noise model (ANM, Equation 1) we can derive that

$$
\frac{\partial}{\partial\theta_{\tau,ij}}\mathbb{E}_{s_{\tau,ij}}[\mathcal{L}_{graph}] = \sigma'(\theta_{\tau,ij})(c_i o_{t,i}(e_{t,j}^2 - e_{t,j}^2)) = \lambda\sigma'(\theta_{\tau,ij}) > 0.
\tag{13}
$$

This is a sigmoidal gradient, whose convergence is analyzed in Section A.1.3. Likewise, we have $\exists\tau \in \{1,...,\tau_{max}\}$, $f_{\phi_j}(\mathcal{X}\odot\boldsymbol{S}_{\tau,ij=1}) \neq f_{\phi_j}(\mathcal{X}\odot\boldsymbol{S}_{\tau,ij=0})$ if time-series $i$ Granger cause $j$, and $\exists\tau$ satisfying

$$
\begin{aligned}
\frac{\partial}{\partial\theta_{\tau,ij}}\mathbb{E}_{s_{\tau,ij}}[\mathcal{L}_{graph}] &= \sigma'(\theta_{\tau,ij})(c_i o_{t,j}((\boldsymbol{x}_{t,j} - f_{\phi_j}(\mathcal{X}\odot\boldsymbol{S}_{\tau,ij=1}))^2 \\
&\quad - (\boldsymbol{x}_{t,j} - f_{\phi_j}(\mathcal{X}\odot\boldsymbol{S}_{\tau,ij=0}))^2) + \lambda).
\end{aligned}
\tag{14}
$$

Assuming that $f_{\phi_j}(\cdot)$ accurately models causal relations in $f_i(\cdot)$ (i.e., DSGNN $f_{\phi_i}$ in Latent data prediction stage model generative function $f_i$ with an error smaller than arbitrarily small value $e_{\mathrm{NN},i}$), applying Equation 1 we have

$$
\begin{aligned}
\frac{\partial}{\partial\theta_{\tau,ij}}\mathbb{E}_{s_{\tau,ij}}[\mathcal{L}_{graph}] &= \sigma'(\theta_{\tau,ij})(c_i o_{t,j}\left(e_{t,j}^2 - (\boldsymbol{x}_{t,j} - f_{\phi_j}(\mathcal{X}\odot\boldsymbol{S}_{\tau,ij=0}))^2\right) + \lambda) \\
&= \sigma'(\theta_{\tau,ij})\left(c_i o_{t,j}(e_{t,j}^2 - (e_{t,j} + \Delta f_{i,j})^2) + \lambda\right) \\
&= \sigma'(\theta_{\tau,ij})(c_i o_{t,j}(-2e_{t,j}\Delta f_{i,j} - \Delta^2 f_{i,j}) + \lambda),
\end{aligned}
\tag{15}
$$

where noise term $\boldsymbol{e}_{t,i} \sim \mathcal{N}(0,\sigma)$, $\Delta f_{i,j} = f_{\phi_j}(\mathcal{X}\odot\boldsymbol{S}_{\tau,ij=1}) - f_{\phi_j}(\mathcal{X}\odot\boldsymbol{S}_{\tau,ij=0})$. This gradient is expected to be negative when $\forall i,j = 1,...,N, \mathbb{E}(c_i\Delta^2 f_{i,j}) \geq p\lambda_0 > \lambda$, where $p$ is the missing probability, i.e., $\mathbb{E}[c_i] = p$ (here we only consider the random missing scenario). Since we can certainly find a $\lambda$ satisfying the above inequality, $\theta_{\tau,ij}$ will go towards $+\infty$ with a properly chosen $\lambda$ and $m_{\tau,ij} \to 1$. Moreover, we show in Appendix Section A.4.6 that CUTS is robust to a wide range of $\lambda$ values. When applies to real data we use Gumbel Softmax estimator for improved performance (Jang et al., 2016).

### A.1.2 THE EFFECTS OF DATA IMPUTATION

To show why data imputation boosts causal discovery, we suppose $\boldsymbol{x}_{t-\tau',j}$, a parent node of $\boldsymbol{x}_{t,i}$ is unobserved and imperfectly imputed with as $\hat{\boldsymbol{x}}_{t-\tau',j} \neq \boldsymbol{x}_{t-\tau',j}$. If time-series $i$ Granger cause $j$, then $f(...,\hat{\boldsymbol{x}}_{t-\tau',j},...) \neq f(...,\boldsymbol{x}_{t-\tau',j},...)$. Let $\delta_{\tau',ij} = f(...,\boldsymbol{x}_{t-\tau',j},...) - f(...,\hat{\boldsymbol{x}}_{t-\tau',j},...)$, and

$$
\begin{aligned}
\frac{\partial}{\partial\theta_{\tau,ij}}\mathbb{E}_{s_{\tau,ij}}[\mathcal{L}_{graph}] &= \sigma'(\theta_{\tau,ij})(c_i o_{t,i}((e_{t,i} + \delta_{\tau',ij})^2 - (e_{t,i} + \delta_{\tau',ij} + \Delta f_{i,j})^2) + \lambda) \\
&= \sigma'(\theta_{\tau,ij})(c_i o_{t,i}(-2(e_{t,i} + \delta_{\tau',ij})\Delta f_{i,j} - \Delta^2 f_{i,j}) + \lambda)
\end{aligned}
\tag{16}
$$

The expectation

$$\mathbb{E}_{e_{t,i}} \left( \frac{\partial}{\partial \theta_{\tau,ij}} \mathbb{E}_{s_{\tau,ij}}[\mathcal{L}_{graph}] \right) = \sigma'(\theta_{\tau,ij})(c_i o_{t,i}(-2\delta_{\tau',ij}\Delta f_{i,j} - \Delta^2 f_{i,j}) + \lambda)$$

As a result, if we cannot find a lower bound for $\delta_{\tau',ij}$, gradient for $\theta_{\tau,ij}$ is not guaranteed to be positive or negative and the true Granger causal relation cannot be recovered. On the other hand, if $x_{t-\tau',j}$ is appropriately imputed with $|\delta_{\tau',ij}| \leq \delta < \lambda_0^2$, we can find $\lambda < p\lambda - p\delta$ to insure negative gradient and $\theta_{\tau,ij}$ will go towards $+\infty$.

### A.1.3 CONVERGENCE OF SIGMOIDAL GRADIENTS

We now analyze the descent algorithm for sigmoidal gradients with learning rate $\alpha$ (for simplicity we denote $\theta_{\tau,ij}$ as $\theta$):

$$\theta_k = \theta_{k-1} + \alpha\lambda\sigma'(\theta_{k-1})$$

This is a monotonic increasing sequence. We show that this sequence converges to $+\infty, \forall \alpha > 0$. If this is not the case, $\exists M > 0$,s.t. $\forall i > 0$, we have $\theta_i \leq M$, since this sequence is monotonic increasing, we have

$$\theta_{k+1} = \theta_k + \alpha\lambda \frac{e^{-\theta_k}}{(1+e^{-\theta_k})^2} \geq \theta_k + \alpha\lambda \frac{e^{-\theta_k}}{(1+e^{-\theta_0})} \geq \theta_k + \alpha\lambda \frac{e^{-M}}{(1+e^{-\theta_0})}$$

then $\exists k$, s.t. $\theta_k > M$, this contradicts with "$\forall i > 0, \theta_i \leq M$", then we have $\theta_k \to +\infty$ and for any finite number $M$, $\theta_k$ can converge to $\geq M$ in finite steps. And likewise sequence $\theta_k = \theta_{k-1} - \alpha\lambda\sigma'(\theta_{k-1})$ converges to $\leq -M$ in finite steps. This enables us to choose a threshold to classify causal and non-causal edges.

### A.2 AN EXAMPLE FOR IRREGULAR TIME-SERIES CAUSAL DISCOVERY

In this section we provide a simple example for irregular causal discovery and show that our algorithm is capable of recovering causal graphs from irregular time-series. Suppose we have a dataset with 3 time-series $\boldsymbol{x}_1, \boldsymbol{x}_2, \boldsymbol{x}_3$, which are generated with

$$x_{t,1} = e_{t,1}, \ x_{t,2} = f_2(x_{t-1,2}) + e_{t,2}, \ x_{t,3} = f_3(x_{t-1,1}, x_{t-1,2}) + e_{t,3}, \tag{17}$$

where $e_1, e_2, e_3$ are the noise terms and follow $\mathcal{N}(0, \sigma)$. We assume only $\boldsymbol{x}_2$ is randomly sampled with missing probability $p_2$

$$o_{t,1} = 1, \ o_{t,2} \sim Ber(1-p_2), \ o_{t,3} = 1, \tag{18}$$

where $Ber(\cdot)$ denotes the Bernoulli distribution. Then the groundtruth causal relations can be illustrated in Figure 3 (left). We use a DSGNN $f_{\phi_2}$ to fit $f_2$ supervised on observed data points of $\boldsymbol{x}_2$, i.e., $\min_{\phi_2} \mathcal{L}_2(\boldsymbol{x}_{t,2}, f_{\phi_2}(\boldsymbol{x}_{t-1,1}))$, $\forall t$, s.t. $\boldsymbol{o}_{t,2} = 1$. Given $f_{\phi_2}$, the unobserved values of $\boldsymbol{x}_2$ can be imputed with $\hat{\boldsymbol{x}}_{t,2} = f_{\phi_2}(\boldsymbol{x}_{t,1})$ and we fit $f_3(\cdot)$ with $f_{\phi_3}(\cdot)$ in *Latent data prediction stage*:

$$\begin{aligned} &\arg\min_{\phi_3} \ \mathcal{L}_2(x_{t,3}, f_{\phi_3}(x_{t-1,1}, \hat{x}_{t-1,2})) \\ =&\arg\min_{\phi_3} \ \mathcal{L}_2(x_{t,3}, f_{\phi_3}(x_{t-1,1}, f_{\phi_2}(x_{t-2,1}))), \end{aligned} \tag{19}$$

and CPGs $\boldsymbol{M}_\tau$ is optimized in *Causal graph fitting stage* with

$$\arg\min_{\boldsymbol{M}_1} \ \mathcal{L}_2(x_{t,3}s_{1,13}, f_{\phi_3}(x_{t-1,1}s_{1,23}, f_{\phi_2}(x_{t-2,1}), x_{t-1,3}s_{1,33})) + \lambda \sum_{i=1}^{3} \sigma(s_{1,i3}), \tag{20}$$

where $s_{1,ij}$ is sampled with Gumbel Softmax technique denoted with Equation 21. Since $x_{t-1,3}$ is invariant to the prediction of $x_{t,3}$ given $x_{t,1}$ and $x_{t,2}$, $s_{1,33}$ can be penalized to zero with a proper $\lambda$. Here we conduct an experiment to verify this example. We set $L = 10000$, random missing probability $p_2 = 0.2$. The illustration of the discovered causal relations is Figure 3. Results show that CUTS without data imputation tends to ignore causal relations from $\boldsymbol{x}_2$ (with missing values) to other time-series. This causal relation $\boldsymbol{x}_2 \to \boldsymbol{x}_3$ are instead "replaced" by $\boldsymbol{x}_3 \to \boldsymbol{x}_3$, which leads to incorrect causal discovery results.

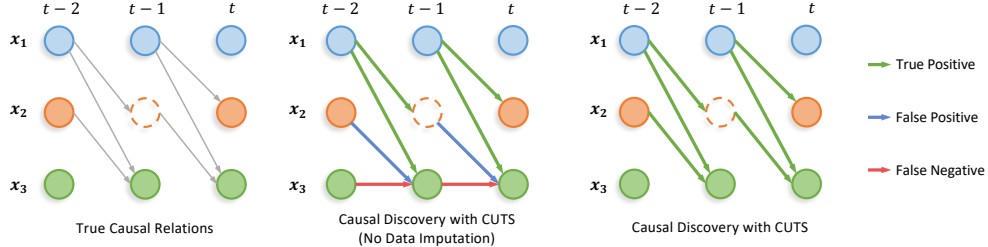

Figure 3: An three-time-series example demonstrating the advantages of introducing data imputation, with the groundtruth causal graph in the left column. The recovered causal graph without data imputation (middle column) shows some false positive and false negative edges, while CUTS (right column) exhibits perfect results.

### A.3 IMPLEMENTATION DETAILS

#### A.3.1 GUMBEL SOFTMAX FOR CAUSAL GRAPH FITTING

In our proposed CUTS, causal relations are modeled with Causal Probability Graph (CPGs), which describe the possibility of Granger causal relations. However, the distributions of CPGs are discrete and cannot be updated directly with neural networks in *Causal graph fitting stage*. To achieve a continuous approximation of the discrete distribution, we leverage Gumbel Softmax technique (Jang et al., 2016), which can be denoted as

$$s_{\tau,ij} = \frac{\exp((\log(m_{\tau,ij}) + g)/\tau)}{\exp((\log(m_{\tau,ij}) + g)/\tau) + \exp((\log(1 - m_{\tau,ij}) + g)/\tau)}, \tag{21}$$

where $g = -\log(-\log(u)), u \sim \text{Uniform}(0, 1)$. The parameter $\tau$ is set according to the "Gumbel tau" item in Table 4. During training we first set a relatively large value of $\tau$ and decrease it slowly.

#### A.3.2 INITIAL DATA FILLING

The missing data points are filled with Zero-Order Holder (ZOH) before the iterative learning process to provide an initial guess $\tilde{x}^{(0)}$. An intuitive solution for initial filling is Linear Interpolation, but it would hamper successive causal discovery. For example, if $x_{t-2,i}$ and $x_{t,i}$ are observed and $x_{t-1,i}$ is missing, $x_{t-1,i}$ is filled as $\tilde{x}^{(0)}_{t-1,i} = \frac{1}{2}(x_{t-2,i} + x_{t,i})$, then $x_{t,i}$ can be directly predicted with $2\tilde{x}^{(0)}_{t-1,i} - x_{t-2,i}$ and other time-series cannot help the prediction of $x_{t,i}$ even if there exists Granger causal relationships. To show the limitation of filling with linear interpolation, we conducted ablation study on VAR datasets with Random Missing ($p = 0.6$). In this experiment, initial data filling with ZOH achieves AUROC of $(0.9766 \pm 0.0074)$ while that with Linear interpolation achieves an inferior accuracy $(0.9636 \pm 0.0145)$. This validates that Zero-order Holder is a better option than linear interpolation as an initial filling implementation.

#### A.3.3 HYPERPARAMETERS SETTINGS

To fit data generation function $f_i$ we use a DSGNN $f_{\phi_i}$ for each time-series $i$. Each DSGNN contains a Multilayer Perceptron (MLP). The layer numbers and hidden layer feature numbers are shown in Table 4. For activation function we use LeakyReLU (with negative slope of 0.05). During training we use Adam optimizer and different learning rate for *Latent data prediction stage* and *Causal graph fitting stage* (shown as "Stage 1 Lr" and "Stage 2 Lr" in Table 4) with learning rate scheduler. The input step for $f_{\phi_i}$ also denotes the chosen max time lag for causal discovery. For VAR and Lorenz-96 datasets we already know the max time lag of the underlying dynamics ($\tau_{max} = 3$), while for NetSim datasets this parameter is chosen empirically.

Table 4: Hyperparameters settings of CUTS in the aforementioned experiments. "$a_1 \rightarrow a_2$" means parameters exponentially increase/decrease from $a_1$ to $a_2$.

| Methods | Hyperparam. | VAR | Lorenz | NetSim | DREAM-3 |
|---|---|---|---|---|---|
| | $n_1$ | 5 | 50 | 200 | 20 |
| | $n_2$ | 15 | 150 | 600 | 30 |
| | $n_3$ | 30 | 300 | 200 | 50 |
| | $\alpha$ | 0.1 | 0.01 | 0.01 | 0.01 |
| | Input step | 3 | 3 | 5 | 5 |
| | Batch size | 128 | 128 | 128 | 128 |
| CUTS | Hidden features | 128 | 128 | 128 | 128 |
| | Network layers | 3 | 3 | 3 | 5 |
| | Weight decay | 0.001 | 0 | 0.001 | 0 |
| | Stage 1 Lr | $10^{-4} \rightarrow 10^{-5}$ | $10^{-4} \rightarrow 10^{-5}$ | $10^{-4} \rightarrow 10^{-5}$ | $10^{-4} \rightarrow 10^{-5}$ |
| | Stage 2 Lr | $10^{-2} \rightarrow 10^{-3}$ | $10^{-2} \rightarrow 10^{-3}$ | $10^{-2} \rightarrow 10^{-3}$ | $10^{-2} \rightarrow 10^{-3}$ |
| | Gumbel $\tau$ | $1 \rightarrow 0.1$ | $1 \rightarrow 0.1$ | $1 \rightarrow 0.1$ | $1 \rightarrow 0.1$ |
| | $\lambda$ | 0.1 | 0.3 | 5 | 5 |

Table 5: Hyperparameters settings of the baseline causal discovery and data imputation algorithms.

| Methods | Hyperparameters | VAR | Lorenz | NetSim | DREAM-3 |
|---|---|---|---|---|---|
| | $\tau_{max}$ | 3 | 3 | 5 | 5 |
| PCMCI | $PC_\alpha$ | 0.05 | 0.05 | 0.05 | 0.05 |
| | CI Test | ParCorr | GPDC | ParCorr | ParCorr |
| | $\mu_1$ | 0.1 | 0.1 | 0.1 | 0.7 |
| eSRU | Learning rate | 0.01 | 0.01 | 0.001 | 0.001 |
| | Batch size | 250 | 250 | 100 | 100 |
| | Epochs | 2000 | 2000 | 2000 | 2000 |
| | Learning rate | 0.05 | 0.05 | 0.05 | 0.05 |
| NGC | $\lambda_{ridge}$ | 0.01 | 0.01 | 0.01 | 0.01 |
| | $\lambda$ Sweeping Range | $0.02 \rightarrow 0.2$ | $0.02 \rightarrow 0.2$ | $0.04 \rightarrow 0.4$ | $0.02 \rightarrow 0.01$ |
| | Epochs | 200 | 200 | 200 | 200 |
| GRIN | Batch size | 128 | 128 | 128 | 128 |
| | Window | 3 | 3 | 3 | 3 |
| | Epochs | 50 | 50 | 50 | 50 |
| LCCM | Batch size | 10 | 10 | 10 | 10 |
| | Hidden size | 20 | 20 | 20 | 20 |
| | Steps | 2000 | 2000 | 2000 | 2000 |
| NGM | Horizon | 5 | 5 | 5 | 5 |
| | GL_reg | 0.05 | 0.05 | 0.05 | 0.05 |
| | Chunk num | 100 | 100 | 100 | 46 |

For baseline algorithm we choose parameters mainly according to the original paper or official repository (PCMCI[3], eSRU[4], NGC[5], GRIN[6]). For fair comparison, we applied parameter searching to determine the key hyperparameters of the baseline algorithms with best performance. Tuned parameters are listed in Table 5.

---

[3]https://github.com/jakobrunge/tigramite
[4]https://github.com/sakhanna/SRU_for_GCI
[5]https://github.com/iancovert/Neural-GC
[6]https://github.com/Graph-Machine-Learning-Group/grin

## A.4 ADDITIONAL EXPERIMENTS

### A.4.1 DREAM-3 EXPERIMENTS

DREAM-3 (Prill et al., 2010) is a gene expression and regulation dataset mentioned in many causal discovery works as quantitative benchmarks (Khanna & Tan, 2020; Tank et al., 2022). This dataset contains 5 models, each representing measurements of 100 gene expression levels. Each measured trajectory has a time length of $T = 21$. This is too low to perform random missing or periodic missing experiments, so with DREAM-3 we only compare our approach with baselines in regular time-series scenarios. The results are shown in Table 11.

### A.4.2 ABLATION STUDY ON VAR AND NETSIM DATASETS

Besides the ablation studies on Lorenz-96 datasets shown in Table 3, we additionally show those on VAR and NetSim in Tables 6 and 7. In Table 6, one can see that "CUTS (Full)" beats other configurations in most scenarios, and the advantage is more obvious with higher missing percentage ($p = 0.6$ for Random Missing and $T_{max} = 4$ for Periodic Missing). On the NetSim datasets with a too small data size $L = 200$, "CUTS (Full)" beats other configurations at a small missing probability ($p = 0.1$).

### A.4.3 ABLATION STUDY FOR EPOCH NUMBERS

In our proposed CUTS, each step can be recognized as a refinement of causal discovery, with builds upon previous imputation results. Since the data imputation and causal discovery mutually boost each other, the performance may be affected by different settings of learning steps. In Table 8 we conduct experiments to show the impact of different epoch numbers on VAR, Lorenz-96, and Netsim datasets. We set $n_1, n_2, n_3$ proportional to original settings.

### A.4.4 PERFORMANCE ON TEMPORAL CAUSAL GRAPH

In the previous experiments, we calculate causal summary graphs with $\tilde{a}_{i,j} = \max\{m_{\tau,ij}\}_{\tau=1}^{\tau_{max}}$, i.e., maximal causal effects along time axis. Our CUTS also supports discovery of 3-dimensional temporal graph $\{m_{\tau,ij}\}$. We conduct experiments to investigate our performance for temporal causal graph discovery. The results are shown in Table 10.

### A.4.5 CAUSAL DISCOVERY WITH STRUCTURED TIME-SERIES DATA

We show in this section that CUTS is able to recover causal relations not only with irregular time-series but also with regular time-series, which is widely used for performance comparison in previous works. We again tested our algorithm on VAR, Lorenz-96, and NetSim datasets, and the results are shown in Table 11. It is observed that our algorithm shows superior performance to baseline methods.

### A.4.6 ROBUSTNESS TO HYPERPARAMETERS SETTINGS

We show that CUTS is robust to changes of hyperparameters settings, with experiment results listed in Table 12. For existing Granger-causality based methods such as NGC (Tank et al., 2022) and eSRU (Khanna & Tan, 2020), parameters $\lambda$ and the maximum time lag $\tau_{max}$ are often required to be tuned precisely. Empirically, $\lambda$ is chosen to balance between the sparsity of the inferred causal relationship and data prediction accuracy, and $\tau_{max}$ is chosen according to the estimated maximum time lag. In this work we find our CUTS gives similar causal discovery results across a wide range of $\lambda$ ($0.01 \sim 0.3$) and $\tau_{max}(3 \sim 9)$.

### A.4.7 LORENZ-96 DATASETS WITH F=40

We further conducted experiments with external forcing constant $F = 40$ on Lorenz-96 datasets instead of $F = 10$ in Section 5.2. We show that our approach produces promising results with $p = 0.3$ for random missing and $T_{max} = 2$ for periodic missing, as shown in Table 13 with AUROC score higher than 0.9.

Table 6: Quantitation results of ablation studies on VAR dataset. "CUTS (Full)" denotes the default settings in this paper. The highest scores (or multiple ones with ignorable gaps) of each column are bolded for clearer illustration.

| Methods | VAR with Random Missing | | VAR with Periodic Missing | |
|---|---|---|---|---|
| | $p = 0.3$ | $p = 0.6$ | $T_{max} = 2$ | $T_{max} = 4$ |
| **CUTS (Full)** | **0.9971 ± 0.0026** | **0.9766 ± 0.0074** | **0.9992 ± 0.0016** | **0.9958 ± 0.0069** |
| ZOH for Imputation | 0.9908 ± 0.0065 | 0.9109 ± 0.0328 | 0.9974 ± 0.0020 | 0.9782 ± 0.0197 |
| GP for Imputation | 0.9964 ± 0.0026 | 0.9240 ± 0.0327 | 0.9980 ± 0.0018 | 0.9442 ± 0.0429 |
| GRIN for Imputation | 0.9963 ± 0.0047 | 0.9014 ± 0.0273 | **0.9992 ± 0.0012** | 0.9818 ± 0.0174 |
| No Imputation | 0.9945 ± 0.0038 | 0.9624 ± 0.0132 | 0.9968 ± 0.0032 | 0.9797 ± 0.0204 |
| Remove CPG for Imput. | **0.9975 ± 0.0020** | 0.9624 ± 0.0132 | 0.9991 ± 0.0016 | 0.9906 ± 0.0123 |
| No Finetuning Stage | 0.9960 ± 0.0073 | 0.9736 ± 0.0074 | 0.9974 ± 0.0032 | 0.9835 ± 0.0160 |

Table 7: Quantitation results of ablation studies on NetSim dataset. "CUTS (Full)" denotes the default settings in this paper.

| Methods | NetSim with Random Missing | |
|---|---|---|
| | $p = 0.1$ | $p = 0.2$ |
| **CUTS (Full)** | **0.7948 ± 0.0381** | 0.7699 ± 0.0550 |
| ZOH for Imputation | 0.7937 ± 0.0349 | 0.7878 ± 0.0361 |
| GP for Imputation | 0.7845 ± 0.0362 | **0.7890 ± 0.0443** |
| GRIN for Imputation | 0.7745 ± 0.0452 | 0.7553 ± 0.0513 |
| No Imputation | 0.7650 ± 0.0272 | 0.7164 ± 0.0343 |
| Remove CPG for Imput. | 0.7912 ± 0.0389 | 0.7878 ± 0.0361 |
| No Finetuning Stage | 0.7650 ± 0.0272 | 0.7164 ± 0.0343 |

### A.4.8    ROBUSTNESS TO NOISE

We experimentally show that CUTS is robust to noise, as shown in Table 9. We choose the non-linear Lorenz-96 datasets for this experiment ($L = 1000, F = 10$) and set additive Gaussian white noise with standard deviation $\sigma = 0.1, 0.3, 1$, respectively.

### A.5    PSEUDOCODE FOR CUTS

We provide the pseudocode of two boosting modules of the proposed CUTS in Algorithm 1 and 2 respectively, and the whole iterative framework in 3. Detailed implementation is provided in supplementary materials and will be uploaded to GitHub soon.

---
**Algorithm 1** Latent data prediction stage

---
**Input:** Time series dataset $\{\boldsymbol{x}_{1:L,1}, ..., \boldsymbol{x}_{1:L,N}\}$; observation mask $\{\boldsymbol{o}_{1:L,1}, ..., \boldsymbol{o}_{1:L,N}\}$;
    Adam optimizer $Adam(\cdot)$
**Output:** DSGNNs parameters $\{\phi_1, ..., \phi_N\}$
    **for** $i = 1$ to $N$ **do**
        $\hat{x}_{t,i} \leftarrow f_{\phi_i}(\boldsymbol{x}_{t-\tau:t-1,i} \odot \boldsymbol{s}_{1:\tau,ij}), \boldsymbol{s}_{\tau,ij} \sim Ber(1 - m_{\tau,ij})$
        $\mathcal{L}_{pred}(\tilde{\mathcal{X}}, \hat{\mathcal{X}}, \mathcal{O}) = \sum_{i=1}^{N} \frac{\langle \mathcal{L}_2(\hat{\boldsymbol{x}}_{1:L,i}, \tilde{\boldsymbol{x}}_{1:L,i}), \boldsymbol{o}_i \rangle}{\frac{1}{L}\langle \boldsymbol{o}_{1:L,i}, \boldsymbol{o}_{1:L,i} \rangle}$
        $\phi_i \leftarrow Adam(\phi_i, \mathcal{L}_{pred})$
    **end for**

---

Table 8: Quantitative comparison on learning step numbers, in terms of AUROC. We set $n_1, n_2, n_3$ proportional to original settings, e.g., if original settings is $n_1 = 50, n_2 = 250, n_3 = 200$ then "50% Steps" means $n_1 = 25, n_2 = 125, n_3 = 100$.

| Methods | VAR with Random Missing | | VAR with Periodic Missing | |
|---|---|---|---|---|
| | $p = 0.3$ | $p = 0.6$ | $T_{max} = 2$ | $T_{max} = 4$ |
| 25% Steps | $0.9912 \pm 0.0041$ | $0.9492 \pm 0.0119$ | $0.9951 \pm 0.0047$ | $0.9818 \pm 0.0158$ |
| 50% Steps | $0.9949 \pm 0.0034$ | $0.9640 \pm 0.0087$ | $0.9978 \pm 0.0028$ | $0.9894 \pm 0.0125$ |
| 75% Steps | $0.9965 \pm 0.0027$ | $0.9729 \pm 0.0075$ | $0.9985 \pm 0.0023$ | $0.9921 \pm 0.0105$ |
| 100% Steps | $0.9971 \pm 0.0026$ | $0.9766 \pm 0.0074$ | $0.9992 \pm 0.0016$ | $0.9958 \pm 0.0069$ |
| Methods | Lorenz-96 with Random Missing | | Lorenz-96 with Periodic Missing | |
| | $p = 0.3$ | $p = 0.6$ | $T_{max} = 2$ | $T_{max} = 4$ |
| 25% Steps | $0.9811 \pm 0.0069$ | $0.9052 \pm 0.0208$ | $0.9924 \pm 0.0050$ | $0.9655 \pm 0.0216$ |
| 50% Steps | $0.9952 \pm 0.0022$ | $0.9456 \pm 0.0153$ | $0.9987 \pm 0.0015$ | $0.9855 \pm 0.0112$ |
| 75% Steps | $0.9987 \pm 0.0016$ | $0.9613 \pm 0.0128$ | $0.9998 \pm 0.0005$ | $0.9930 \pm 0.0062$ |
| 100% Steps | $0.9996 \pm 0.0005$ | $0.9705 \pm 0.0118$ | $1.0000 \pm 0.0000$ | $0.9959 \pm 0.0042$ |
| Methods | NetSim with Random Missing | | | |
| | $p = 0.1$ | | $p = 0.2$ | |
| 25% Steps | $0.7737 \pm 0.0346$ | | $0.7403 \pm 0.0355$ | |
| 50% Steps | $0.7963 \pm 0.0399$ | | $0.7699 \pm 0.0443$ | |
| 75% Steps | $0.7961 \pm 0.0390$ | | $0.7714 \pm 0.0503$ | |
| 100% Steps | $0.7948 \pm 0.0381$ | | $0.7699 \pm 0.0550$ | |

Table 9: Accuracy of CUTS on Lorenz-96 datasets with different noise levels. The accuracy is calculated in terms of AUROC.

| Methods | Noise $\sigma$ | Lorenz-96 with Random Missing | |
|---|---|---|---|
| | | $p = 0.3$ | $p = 0.6$ |
| | 0.1 | $1.0000 \pm 0.0000$ | $0.9843 \pm 0.0073$ |
| **CUTS** | 0.3 | $1.0000 \pm 0.0001$ | $0.9825 \pm 0.0080$ |
| | 1 | $0.9999 \pm 0.0002$ | $0.9722 \pm 0.0108$ |

## A.6 MSE CURVE FOR DATA IMPUTATION

The Mean Square Error (MSE) of the imputed time-series, imputed time-series without the help of causal graph, and the groundtruth time-series during the whole training process are shown in Figure 4. We can see that under all configurations our approach successfully imputes missing values with significantly lower MSE compared to initially filled values. Furthermore, in most settings imputing time-series without the help of causal graph are prone to overfit. The imputed time-series then boost the subsequent causal discovery module, and discovered causal graph help to prevent overfit in imputation.

Table 10: Quantitative comparison for 3-dimensional temporal causal graph discovery on VAR datasets, in terms of AUROC.

| Methods | VAR with Random Missing | | |
| --- | --- | --- | --- |
| | $p = 0$ | $p = 0.3$ | $p = 0.6$ |
| CUTS | $0.9979 \pm 0.0018$ | $0.9848 \pm 0.0053$ | $0.9170 \pm 0.0127$ |
| Methods | VAR with Periodic Missing | | |
| | $T_{max} = 1$ | $T_{max} = 2$ | $T_{max} = 4$ |
| CUTS | $0.9973 \pm 0.0024$ | $0.9938 \pm 0.0036$ | $0.9612 \pm 0.0286$ |

Table 11: Accuracy of CUTS and five other baseline causal discovery algorithms on VAR, Lorenz-96, NetSim, and DREAM-3 datasets without missing values. The accuracy is calculated in terms of AUROC.

| Methods | Lorenz-96 | VAR | NetSim | DREAM-3 |
| --- | --- | --- | --- | --- |
| PCMCI | $0.7515 \pm 0.0381$ | $0.9999 \pm 0.0002$ | $0.7692 \pm 0.0414$ | $0.5517 \pm 0.0261$ |
| NGC | $0.9967 \pm 0.0058$ | $0.9988 \pm 0.0015$ | $0.7616 \pm 0.0504$ | $0.5579 \pm 0.0313$ |
| eSRU | $0.9996 \pm 0.0005$ | $0.9949 \pm 0.0040$ | $0.6817 \pm 0.0263$ | $0.5587 \pm 0.0335$ |
| LCCM | $0.9967 \pm 0.0058$ | $0.9988 \pm 0.0015$ | $0.7616 \pm 0.0504$ | $0.5046 \pm 0.0318$ |
| NGM | $0.9996 \pm 0.0005$ | $0.9949 \pm 0.0040$ | $0.6817 \pm 0.0263$ | $0.5477 \pm 0.0252$ |
| **CUTS** | $\mathbf{1.0000 \pm 0.0000}$ | $\mathbf{0.9999 \pm 0.0002}$ | $\mathbf{0.8277 \pm 0.0435}$ | $\mathbf{0.5915 \pm 0.0344}$ |

Table 12: Accuracy of causal discovery results of CUTS under different hyperparameters $\lambda$ and $\tau_{max}$ settings.

| | $\lambda$ | AUROC | $\tau_{max}$ | AUROC |
| --- | --- | --- | --- | --- |
| | 0.01 | $0.9962 \pm 0.0029$ | 3 | $0.9971 \pm 0.0026$ |
| CUTS | 0.03 | $0.9964 \pm 0.0029$ | 6 | $0.9972 \pm 0.0032$ |
| | 0.1 | $0.9971 \pm 0.0026$ | 9 | $0.9972 \pm 0.0042$ |
| | 0.3 | $0.9962 \pm 0.0027$ | | |

Table 13: Comparison of CUTS with (i) PCMCI, eSRU, NGC combined with imputation method ZOH, GP, GRIN and (ii) LCCM, NGM which does not need data imputation. Results are averaged over 4 randomly generated datasets.

| Method | Imputation | Random Missing | Periodic Missing |
| --- | --- | --- | --- |
| | | $p = 0.3$ | $T_{max} = 2$ |
| | ZOH | $0.7995 \pm 0.0361$ | $0.8164 \pm 0.0313$ |
| PCMCI | GP | $0.8124 \pm 0.0221$ | $0.7871 \pm 0.0323$ |
| | GRIN | $0.8193 \pm 0.0329$ | $0.7816 \pm 0.0361$ |
| | ZOH | $0.8067 \pm 0.0267$ | $0.8558 \pm 0.0248$ |
| NGC | GP | $0.8350 \pm 0.0314$ | $0.8250 \pm 0.0257$ |
| | GRIN | $0.6293 \pm 0.0523$ | $0.7114 \pm 0.0129$ |
| | ZOH | $0.8883 \pm 0.0131$ | $0.9463 \pm 0.0208$ |
| eSRU | GP | $0.9499 \pm 0.0061$ | $0.8893 \pm 0.0160$ |
| | GRIN | $0.9417 \pm 0.0199$ | $\mathbf{0.9494 \pm 0.0129}$ |
| LCCM | | $0.6437 \pm 0.0267$ | $0.6215 \pm 0.0343$ |
| NGM | | $0.6734 \pm 0.0403$ | $0.7522 \pm 0.0520$ |
| **CUTS (Proposed)** | | $\mathbf{0.9737 \pm 0.0105}$ | $0.9289 \pm 0.0145$ |

---

**Algorithm 2** Causal graph fitting stage

---

**Input:** Time series dataset $\{\boldsymbol{x}_{1:L,1}, ..., \boldsymbol{x}_{1:L,N}\}$; observation mask $\{\boldsymbol{o}_{1:L,1}, ..., \boldsymbol{o}_{1:L,N}\}$; Adam optimizer $Adam(\cdot)$; Gumbel softmax function $Gumbel(\cdot)$ described with Equation 21

**Output:** Causal probability $m_{\tau,ji}, \forall j = 1, ..., N$

    **for** $i = 1$ to $N$ **do**

        $\hat{x}_{t,i} \leftarrow f_{\phi_i}(\boldsymbol{x}_{t-\tau:t-1,i} \odot \boldsymbol{s}_{1:\tau,ij}), \boldsymbol{s}_{\tau,ij} = Gumbel(1 - m_{\tau,ij})$

        $\mathcal{L}_{graph}(\tilde{\mathcal{X}}, \hat{\mathcal{X}}, \mathcal{O}, \boldsymbol{\theta}) = \mathcal{L}_{pred}(\tilde{\mathcal{X}}, \hat{\mathcal{X}}, \mathcal{O}) + \lambda||\sigma(\boldsymbol{\theta})||_1$

        $\boldsymbol{\theta} \leftarrow Adam(\boldsymbol{\theta}, \mathcal{L}_{graph})$

    **end for**

---

**Algorithm 3** Causal Discovery from Irregular Time-series (CUTS)

---

**Input:** Time series dataset $\{\boldsymbol{x}_{1:L,1}, ..., \boldsymbol{x}_{1:L,N}\}$ with time-series length $L$; observation mask $\{\boldsymbol{o}_{1:L,1}, ..., \boldsymbol{o}_{1:L,N}\}$; Zero-order holder (ZOH) imputation algorithm $ZOH(\cdot)$; Adam optimizer $Adam(\cdot)$

**Output:** Discovered causal graph

    Initialize $\tilde{\boldsymbol{x}}_{1:L,i}^{(0)} = ZOH(x_{1:L,i})$, Causal Probability Graphs $\boldsymbol{M}_\tau = \boldsymbol{0}, \forall \tau = 1, ..., \tau_{max}$

    # Warming up

    **for** $n_1$ iterations **do**

        Update $\{\phi_1, ..., \phi_N\}$ with Algorithm 1

        Update $\boldsymbol{M}_\tau$ with Algorithm 2

    **end for**

    # Causal discovery with data imputation

    **for** $n_2$ iterations **do**

        Update $\{\phi_1, ..., \phi_N\}$ with Algorithm 1

        Update $\boldsymbol{M}_\tau$ with Algorithm 2

        **for** $i = 1$ to $N$ **do**

            Data update: $\tilde{\boldsymbol{x}}_{t,i}^{(m+1)} \leftarrow \begin{cases} (1-\alpha)\tilde{\boldsymbol{x}}_{t,i}^{(m)} + \alpha\hat{\boldsymbol{x}}_{t,i}^{(m)} & o_{t,i} = 0 \\ \tilde{\boldsymbol{x}}_{t,i}^{(0)} & o_{t,i} = 1 \end{cases}$

        **end for**

    **end for**

    # Finetuning

    Reset $o_{t,i} \leftarrow 1, \forall t = 1, ..., T, i = 1, ..., N$

    **for** $n_3$ iterations **do**

        Update $\{\phi_1, ..., \phi_N\}$ with Algorithm 1

        Update $\boldsymbol{M}_\tau$ with Algorithm 2

    **end for**

    **for** $i = 1$ to $N$ **do**

        **for** $j = 1$ to $N$ **do**

            $\tilde{a}_{i,j} = \max\left(m_{1,ij}, ..., m_{\tau_{max},ij}\right)$

        **end for**

    **end for**

    **return** Discovered causal adjacency matrix $\hat{\boldsymbol{A}}$ where each elements is $\tilde{a}_{i,j}$.

---

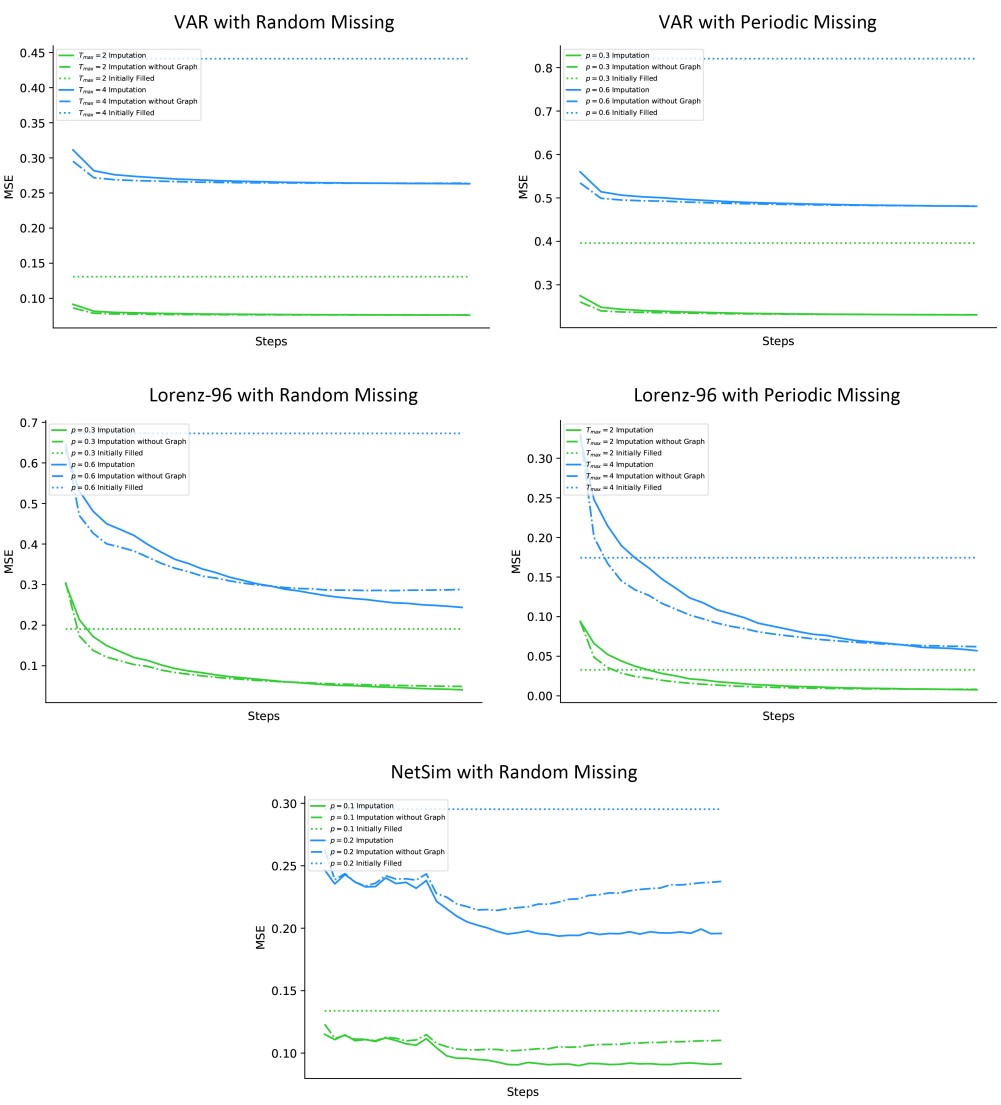

Figure 4: Average MSE curve of imputed data on VAR datasets with Random Missing / Periodic Missing (top), Lorenz-96 datasets under Random Missing / Periodic Missing (middle), and NetSim datasets with Random Missing (bottom).

