# OpenReview forum: "CUTS: Neural Causal Discovery from Irregular Time-Series Data"
_ICLR.cc/2023/Conference — ICLR 2023 poster_

### Official Review · Reviewer_jtS8 · 2022-10-23

**Confidence:** 4
**Correctness:** 2
**Technical Novelty And Significance:** 2
**Empirical Novelty And Significance:** 2
**Recommendation:** 6

**Clarity, Quality, Novelty And Reproducibility:**

In general, this paper is written clearly apart from the "periodic missing" part and the idea is easy to follow. Empirically, I prefer more real-world experiments but the ablation study here is also useful to demonstrate the model properties. As for reproducibility, I wonder how the hyper-parameters are tuned. Is it based on a validation dataset and tuned with MSE error? What is the data splitting ratio?

For novelty, I think the author should include the citation of [2], which also tackles the missing value and structure learning at the same time. The formulation is very similar. Overall, I think it is not a very novel model, and similar ideas have been explored in static settings.

**Strength And Weaknesses:**

## Strength
Granger causality with deep learning has been a popular research topic in recent years, and this model is an extension of some well-known previous work. The idea behind this model is intuitive and easy to follow. Despite some typos in the math, the model formulation is easy to understand. Although I think the author should conduct more real-world experiments (e.g. DREAM3/4 dataset), the author has conducted sufficient ablation studies for synthetic experiments to demonstrate some of the properties of the model.

## Weakness
After reading the paper, I have some concerns regarding the proposed method: (1) theoretical results and soundness; (2) clarity regarding the periodic missing.

Specifically, first, I am not fully sure the proposed objective in eq.10 is correct. Since the adjacency matrix is sampled from an independent Bernoulli distribution, which is modeled by the parameter $\theta$. This means this is very similar to a variational framework, where the Bernoulli distribution is a mean-field approximation, see the formulation [1] and [2] for details. This means the with Gaussian noise variable, the $L_{pred}$ in eq.10 is similar to the likelihood term in ELBO and $\Vert \sigma(\theta) \Vert_1$ is similar to the prior, however, the entropy of the Bernoulli distribution is missed in eq.10. This means the proposed objective is not a proper likelihood, so how it can guarantee it recovers the ground truth graph? (I will explain my concern of theorem 1 in the following). How to explain the differences?

The author provides theorem 1 to justify the convergence of the CUTs. However, I am not fully sure this will hold. For equation 15, the inputs to $f_{\phi_i}$ should be the parents of node $x_{t,i}$, why it is $x_{t-\tau:t-1, i}$? Is this a typo?
In eq.17, when $S$ and $f_\phi$ are not the ground truth, how this two cases with $S_{ij=1}$ equals $S_{ij=0}$? Also, the negative gradient has a derivative of the sigmoid function, meaning that the magnitude of the gradient will go toward 0, meaning that more rigorous analysis is needed to claim convergence.

For equations 18 and 19, I am not sure the assumption "$f_\phi$ accurately models the $f_i$" is a reasonable one. Without learning a good graph, it is hard to learn a good $f_\phi$. Thus, I don't think this assumption will hold in general. In addition, due to a similar reason as above, careful analysis is needed to claim convergence.


For the clarity, if I understand correctly, the author tries to model the non-uniform sampling interval as the missing value problem (i.e. periodic missing). In this case, it can only tackle a subset of "non-uniform sampling interval" since it assumes a fixed period. If that is the case, I wonder how CUTs will work in the following example: $x_1$ is sampled with interval T and $x_2$ is sampled with interval 2T, In that case, $x_2$ will be missing at $t=T, 3T, 5T, ...$ (assuming start t=0). So how CUTs can impute this value $x_2$ in-between since no observation is provided?

[1] Geffner, T., Antoran, J., Foster, A., Gong, W., Ma, C., Kiciman, E., ... & Zhang, C. (2022). Deep End-to-end Causal Inference. arXiv preprint arXiv:2202.02195.
[2] Morales-Alvarez, P., Lamb, A., Woodhead, S., Jones, S. P., Allamanis, M., & Zhang, C. (2021). VICause: Simultaneous Missing Value Imputation and Causal Discovery with Groups. arXiv preprint arXiv:2110.08223.


**Summary Of The Paper:**

This paper proposes a deep learning-based Granger causality model for discovering the Granger causal graph and imputing missing values at the same time. The author claim this model can handle missing completely at random and periodic missing scenarios.
The main methodology is to iteratively apply the data imputation and graph learning, where the graph learning stage is further split into three sub-stages: (1) without imputed value; (2) with imputed value but not as model supervisory; (3) with imputed value.
The graph is sampled from an independent Bernoulli distribution with a Gumbel-softmax gradient estimator.

Empirically, the author conducts 2 synthetic experiments with the Netsim fMRI dataset, where the model achieves better results.

**Summary Of The Review:**

My main concern regarding this paper is the technical soundness and claims made by the author. In general, I think this paper can be improved if the author can clarify the concerns and elaborate more on its novelty.

---

> ### Author Response · Authors · 2022-11-17
> **Point-by-point Response to Reviewer jtS8 (Part 1/2)**
>
>
> Thank you for taking time to review this work. Your suggestions are of great help to us and we amend our paper accordingly. Below are the point-to-point responses.
>
> > I think the author should conduct more real-world experiments (e.g. DREAM3/4 dataset)
>
> Thank you for this valuable suggestion. The validation on real data is indeed important and we conducted experiments on DREAM-3 datasets following your advice, as shown in Appendix Section A.4.1.
>
> > This means this is very similar to a variational framework, where the Bernoulli distribution is a mean-field approximation, see the formulation [1] and [2] for details.
>
> We agree that both frameworks jointly conduct missing data imputation and causal graph discovery, however, we would like to argue that our approach is quite different and have many potential advantages.
> 1. Existing variational-based approaches are with ELBO objective, and can not guarantee convergence. Instead, we use explicit non-probabilistic dynamic modeling of causal effects and is proven to be with convergence guarantee given appropriate assumptions (Theorem 1).
> 2. Variational framework conduct causal discovery from the probabilistic view, while Granger-causality is more closely related to functional causal relationship and it is thus more effective to build functional causal models.
>
> > For equation 15, the inputs to $f_{\phi_i}$ should be the parents of node $x_{t,i}$ , why it is $x_{t-\tau:t-1, i}$? Is this a typo?
>
> Yes, the input to $f_{\phi_i}$ is indeed the parents, i.e., $X\odot S$, and we have corrected this typo in the revised manuscript. Thank you very much.
>
> > In eq.17, when $S$ and $f_\phi$ are not the ground truth, how this two cases with $S_{ij=1}$ equals $S_{ij=0}$? ... Without learning a good graph, it is hard to learn a good $f_{\phi}$. Thus, I don't think this assumption will hold in general.
>
> Firstly, according to the definition of nonlinear Granger causality [1], if time-series $i$ does not Granger cause $j$, then
>
> $\forall x_{t-\tau:t-1,j}$ and $\forall x_{t-\tau:t-1,j}' \neq x_{t-\tau:t-1,j}$, we have $ f_j(X)=f_j(X')$
>
> so even if $S$ is not the ground truth we still have $f_i(X \odot S_{ij=1})=f_i(X'\odot S_{ij=0})$ (we updated definition 1 for better clarity).
>
> Secondly, the assumption that $f_{\phi_i}$ accurately models $f_i$ is fair both theoretically and empirically, as detailed as follows:
> 1. According to the universal approximation theorem [2], it is fair to assume that our DSGNN can learn a good $f_{\phi_i}$ with large network width and sufficient data.
> 2. Empirically, even without a causal graph, it is possible to approximate $f_i$ with $f_{\phi_i}$ at high accuracy, as verified by many neural-network-based time-series imputation algorithms [3][4].
>
> [1] Tank, A., Covert, I., Foti, N., Shojaie, A., & Fox, E. B. (2022). Neural granger causality. IEEE Transactions on Pattern Analysis and Machine Intelligence, 44(8), 4267–4279.
>
> [2] Hornik, K., Stinchcombe, M., & White, H. (1989). Multilayer feedforward networks are universal approximators. Neural Networks, 2(5), 359–366.
>
> [3] Cao, W., Wang, D., Li, J., Zhou, H., Li, L., & Li, Y. (2018). BRITS: Bidirectional recurrent imputation for time series. Advances in Neural Information Processing Systems, 31.
>
> [4] Luo, Y., Cai, X., ZHANG, Y., Xu, J., & xiaojie, Y. (2018). Multivariate Time Series Imputation with Generative Adversarial Networks. Advances in Neural Information Processing Systems, 31.

---

> ### Author Response · Authors · 2022-11-17
> **Point-by-point Response to Reviewer jtS8 (Part 2/2)**
>
>
> > Also, the negative gradient has a derivative of the sigmoid function, meaning that the magnitude of the gradient will go toward 0, meaning that more rigorous analysis is needed to claim convergence.
>
> We clarify this issue in Section A.1.3 of the revised manuscript. Although the gradient goes toward 0, we prove that $\forall M$, the parameter $\theta$ converges to $\geq M$ in finite steps if there exist Granger causal relationships. This is sufficient for recovering the true causal graphs.
>
> > For the clarity, if I understand correctly, the author tries to model the non-uniform sampling interval as the missing value problem (i.e. periodic missing). In this case, it can only tackle a subset of "non-uniform sampling interval" since it assumes a fixed period
>
> Yes, we address both "periodic missing" and  "random missing" in this paper, and the latter can model the time series with non-uniform sampling intervals.
>
> > If that is the case, I wonder how CUTs will work in the following example: ...
>
> We deeply thank you for this question.
> In this example, $x_{2,t}$ at $t=T,3T,...$ is imputed in the following ways: (i) when its causes (or parents) are observed, the missing values can be generated from the functional model $x_{2,t} = f_2(pa_{x_{2,t}})$; (ii) when the parents are not fully observed which often happens at high missing percentage, we perform imputation from its observed indirect ascendents. This is well validated in our "periodic missing" experiments, even with $T_{max}=4$ or $p=0.6$ Our approach still performs excellently.
>
>
> > For novelty, I think the author should include the citation of [2], which also tackles the missing value and structure learning at the same time. The formulation is very similar. Overall, I think it is not a very novel model, and similar ideas have been explored in static settings.
>
> Thank you for the suggestion. We added citations to the referred paper and clarified the difference in the related works section of the revised version.
>
> We respectfully disagree with the judgment on the novelty of our approach. Although also developed under the joint optimization scheme, existing approaches working in static settings, such as VICause and DECI, cannot be applied to time series data and are largely different from ours. Specifically, these approaches are variational-based with ELBO objective and have no convergence guarantee, while we use explicit non-probabilistic dynamic modeling of causal effects and provide convergence guarantee given appropriate assumptions (Theorem 1).
>
> > As for reproducibility, I wonder how the hyper-parameters are tuned. Is it based on a validation dataset and tuned with MSE error? What is the data splitting ratio?
>
> The hyper-parameters are tuned with grid search on validation dataset (independently generated with different random seeds and the same size). We did the same for baseline algorithms to maintain fairness. The detailed descriptions are listed in Section A.3.3.
>
>
> Thanks again for your valuable insights!

---

> > ### Comment · Reviewer_jtS8 · 2022-12-02
> > **Reply**
> >
> > Thanks for the author's response, it addressed most of my concerns. I have increased my score. One additional comment: the Dream3 performance seems a bit low compared to the reported number in the literature, e.g. eSRU. What causes the difference?

---

> > > ### Author Response · Authors · 2022-12-03
> > > **Dream3 Performance**
> > >
> > > Thanks for your appreciation! This means a lot to us.
> > >
> > > > the Dream3 performance seems a bit low compared to the reported number in the literature, e.g. eSRU. What causes the difference?
> > >
> > > The performance of our CUTS are robust to a wide range of hyper-parameters settings (Section A.4.6). So they can be easily tuned. However this may not be the case for baseline methods, e.g., eSRU.
> > >
> > > According to the original paper, the eSRU performance on Dream3 (AUROC) is evaluated by **sweeping through different values** of the regularization parameter $\lambda_1$. However, in the eSRU paper and code repository, the author only provide a range for this parameter, i.e. [0.1, 3.162]. As the result, there are infinite groups of values to sweep through. Even with a grid search, we cannot guarantee to find the setting to obtain optimal results.
> > >
> > > In experiments, with significant effort we still cannot get similar results to the original eSRU paper, this also really puzzles us. We are still investigating this problem. The new results will be updated in the final version if this paper is accepted.

---

### Official Review · Reviewer_ADkU · 2022-10-27

**Confidence:** 3
**Correctness:** 3
**Technical Novelty And Significance:** 2
**Empirical Novelty And Significance:** 3
**Recommendation:** 5

**Clarity, Quality, Novelty And Reproducibility:**

Paper is reasonably clear and the authors have described the algorithms in good detail in the Appendix also.  I believe the work can be reproduced with significant effort (due to multiple algorithms involved).

**Strength And Weaknesses:**

**Strengths**

- Problem of causal discovery from unstructured time series data is an important problem and the paper presents a framework which is intuitive and relatively easy to follow.

- The authors have put in significant effort to benchmark their technique with various combinations of causal discovery and missing pattern imputation techniques.

**Weakness**
- The paper as a whole does not do a good job in providing a technically rigorous narrative on why or how Data imputation boost Causal Discovery.  The vice versa argument seems reasonable enough, but given that the imputation is just confined to addressing two patterns in time series data (random and periodic patterns), I wonder if the causal discovery would benefit with the limited insight provided by the data imputation framework.
- To support these claims, I do notice that in general the AUROC improvements for VAR with Random missing are marginal compared to other competing methods in Table 1.

**Minor writing issues**

- ParCorr first occurrence is not defined on Page 6.
- ZOH first occurrence is not defined on Page 5.
- Table 2 title should be Quantitative and Table 3 title should also be Quantitative

**Summary Of The Paper:**

In this paper, the authors present a Granger causal discovery framework for unstructured time series data which leverages an alternating formulation comprising of a) data imputation and b) subsequent causal discovery. The basic premise is centered around providing empirical results on how data imputation aids the causal discovery and vice versa. Ablation studies based results along with comparisons using different combinations of Causal Discovery and data imputation techniques (Random and periodic missing patterns only considered) indicate the superior performance of the proposed CUTS method.

**Summary Of The Review:**

I believe the paper presents a good empirical study of a framework which nicely combines two techniques a) data imputation and b) causal discovery in an alternating style framework to deal with the problem of causal discovery from unstructured time series data. While the approach is interesting and the paper is reasonably easy to follow, the results for some cases do not indicate significant improvements. However, the more major concern I have is with the technical rigor is missing to support some claims on data imputation supporting causal discovery.

---

> ### Author Response · Authors · 2022-11-17
> **Point-by-point Response to Reviewer ADkU**
>
>
> Thank you! Your suggestions are of great help to us and we revised our paper accordingly. Below are the point-to-point responses.
>
> > The paper as a whole does not do a good job in providing a technically rigorous narrative on why or how Data imputation boost Causal Discovery. The vice versa argument seems reasonable enough, but given that the imputation is just confined to addressing two patterns in time series data (random and periodic patterns), I wonder if the causal discovery would benefit with the limited insight provided by the data imputation framework. To support these claims, I do notice that in general the AUROC improvements for VAR with Random missing are marginal compared to other competing methods in Table 1.
>
> Thank you for this suggestion! This suggestion is of great help to us. In the revised manuscript, we provide both theoretical explanation and experimental validation about how data imputation boosts causal discovery:
> 1. Mathematically, we showed that without properly imputed data points, the causal graph edge $\theta_{t,ij}$ does not converge to $+\infty$ even if there exists Granger causal relationship. On the contrary, with a limited imputation error, we can still find some $\lambda$ with convergence guarantee. The proof is added in section A.1.2.
> 2. As for the experiments, validation is provided in both ablation studies and performance comparisons with state-of-the-arts.  (i) In the ablation study section (describe in section 5.4 and A4.2, results shown in Tab. 3, 6, 7) we can see that data imputation boosts causal discovery with significant improvement.
> (ii) As for quantitative comparison with existing methods, results are shown in Section 5. We can see that after imputation, AUC is largely raised with more missing data and our approach provides slight improvement but near perfect results with fewer missing data. The detailed figures are shown as follows:
>
> The results on 6/10 settings with large percentage missing data, with the increment compared to the second competitor as
> | Setting | Increments of AUC |
> |:--------------------------:|:-----------:|
> | VAR with $p=0.6$ | **2.83%** |
> | VAR with $T_{max}=4$ | **1.74%** |
> | Lorenz-96 with $p=0.6$ | **1.13%** |
> | Lorenz-96 with $T_{max}=4$ | **1.47%** |
> | Netsim with $p=0.1$ | **3.81%** |
> | NetSim with $p=0.2$ | **0.40%** |
>
> The improvement of AUC score on 4/10 cases with few missing entries (VAR and Lorenz-96 with $p=0.3$ and $T_{max}=2$) is indeed relatively small, however, the AUC is very close to 1 (e.g. 0.9996 for Lorenz-96 with $p=0.3$) and even small improvement on such a high AUC is challenging and a big step towards perfection.
>
> > Minor writing issues
>
> Thank you for the suggestion, and have amended the manuscript accordingly. We added descriptions of ParCorr in section 5 and define the full name of ZOH in section 4.1.

---

### Official Review · Reviewer_SHw1 · 2022-10-27

**Confidence:** 3
**Correctness:** 3
**Technical Novelty And Significance:** 2
**Empirical Novelty And Significance:** 2
**Recommendation:** 6

**Clarity, Quality, Novelty And Reproducibility:**

The paper is generally well written and easy to follow. The novelty of the method is limited in light of the papers mentioned [1-4]. Most relevant details are included for reproducibility.

**Strength And Weaknesses:**

Strengths:
- The paper tackles the important problem of modelling timeseries data with missing entries.
- The experimental results show competitive or superior results to relevant baselines.
- The proposed method seems to incorporate a lot of small tweaks that improve the model performance - there seems to be value in analysing the training behaviour a bit further.

Weaknesses:
- Some important related works are missing: [1] tackles temporal causal discovery with Neural ODEs that would be able to handle inconistent sampling intervals, [2] performs joint structure learning and data imputation, [3] performs temporal causal discovery using the NOTEARS framework for continuous DAG learning. Methods based on the same framework have been applied to static data for joint causal discovery and data imputation [4]. All these weaken the novelty of this paper.
- The paper mentions that it is based on Granger causality. However, the current formulation also allows for an interpretation as an additive noise model: ie $x_i = f(pa_i) + e_i$. Could you please comment on this? This interpretation would also allow for the identification of the temporal causal graph $A_{0, \tau}$ rather than just the summary graph $\hat{A} = max_t A_{t}$.
  - Please add some comments about the difference between Granger causality and ANMs or PCMCI that also identify the temporal causal graph.
- It is unclear how $\tau_{max}$ is chosen. Is this assumed to be known? What if this isn't known?
- All experiments use missing data. It would be great to see a baseline comparing to datasets with full observability.

Misc:
- What's the intuition of using the moving average as a training signal for the imputation network?
- For the graph discovery stage - do you calculate an expectation over multiple graph samples or is this amortised over different batches? Or do you use the same graph sample for optimising this loss?
- Please explain ZOH earlier in the text.
- Eq 3: what is $e$? What's the assumption about it? This might make or break the use of the L2 loss.
- Please pay attention to the use of `\citep` and `\citet`.
- Eq 5: You use inconsistent $\tau=0...$ and $\tau=1,...$.
- p5 just above eq 11: I believe $n_2$ should be $n_3$.
- p2: " to conduct causal inference and .." - should this be "causal discovery"? Causal inference tackles the question of inferring causal estimates (e.g. ATEs).

[1] Bellot, Alexis, Kim Branson, and Mihaela van der Schaar. "Neural graphical modelling in continuous-time: consistency guarantees and algorithms." International Conference on Learning Representations. 2021.
[2] Morales-Alvarez, Pablo, et al. "VICause: Simultaneous Missing Value Imputation and Causal Discovery with Groups." arXiv preprint arXiv:2110.08223 (2021).
[3] Pamfil, Roxana, et al. "Dynotears: Structure learning from time-series data." International Conference on Artificial Intelligence and Statistics. PMLR, 2020.
[4] Geffner, Tomas, et al. "Deep End-to-end Causal Inference." arXiv preprint arXiv:2202.02195 (2022).

**Summary Of The Paper:**

The paper presents an auto-regressive style model for joint causal graph learning and missing data imputation for unstructured temporal data. The model is trained in a two-step procedure alternating between training the imputation model and the graph learning (EM-style algorithm). The model parametrises the graph using an independent Bernoulli distribution and the Gumbel softmax for enabling gradient based learning. The paper compares the method to relevant temporal discovery methods and achieves competitive performance.

**Summary Of The Review:**

The paper tackles and important and interesting problem of discovering causal graphs from temporal data with missing values. However, the evaluation only compares to baselines without proper data imputation capabilities and does not offer a comparison on fully observed data. Furthermore, important literature is missing, and it would be good to discuss some of the model assumptions around Granger causality vs other causal discovery assumptions. Lastly, the model choices about noise distributions and loss functions should be better justified.

---

> ### Author Response · Authors · 2022-11-17
> **Point-by-point Response to Reviewer SHw1 (Part 1/2)**
>
>
> Thank you for taking time to review this work. Your suggestions are of great help to us and we amend our paper accordingly. Below are the point-to-point responses.
>
> > Some important related works are missing... All these weaken the novelty of this paper.
>
> Thank you for the suggestion. We have added citations to these papers in the related works section and added experimental comparisons with two most related ones NGM [1] and LCCM [2] (mentioned by another reviewer).
>
> [1] Bellot, A., Branson, K., & Schaar, M. van der. (2022, February 3). Neural graphical modelling in continuous-time: Consistency guarantees and algorithms. International Conference on Learning Representations.
> [2] Brouwer, E. D., Arany, A., Simm, J., & Moreau, Y. (2021, March 17). Latent Convergent Cross Mapping. International Conference on Learning Representations.
>
> However, we respectfully disagree that these works weaken our novelty, since our approach is largely different from and advantageous over these works, as clarified in the following:
>
>  1. Although some work such as NGM and LCCM also conduct causal discovery with irregular/incomplete time series, these methods built on Neural ODEs tend to degenerate on non-ODE data (e.g. VAR), and do not model the causal connections on multiple time lag $\tau$ either. Differently, our approach displays superior performance on diverse datasets (either ODE or non-ODE), and explicitly discovers temporal causal graphs.
>
>  2. Existing methods for joint imputation and causal discovery, such as VICause and DECI, are limited to static data, while our framework conducts explicit dynamic modeling and enables imputing irregular time series. Besides, these two approaches are variational-based with ELBO objective, and have no convergence guarantee. Instead, we use explicit non-probabilistic dynamic modeling of causal effects and provide convergence guarantee given appropriate assumptions (Theorem 1).
>
>  3. Besides data imputation and causal discovery, combining neural networks and Granger causality would also benefit the explanability and robustness of other diverse downstream tasks (feature selection, prediction, classification, etc.).  We strongly believe this approach has potential and would benefit the whole machine learning community.
>
> > The paper mentions that it is based on Granger causality. However, the current formulation also allows for an interpretation as an additive noise model: ie $x_i=f(pa_i)+e_i$. Could you please comment on this? This interpretation would also allow for the identification of the temporal causal graph $A_{0,\tau}$ rather than just the summary graph $\hat{A}=max_t A_t$.
>
> This is an interesting opinion and thanks for that. Yes, the current formulation does allow for additive noise models (ANMs). In the revised manuscript, this is included in our formulation (Eq. 1). We also include the ANM models in the proof of Theorem 1 (Appendix Section A.1), which seems to work well with our approach.
>
> Our formulation does support inferring temporal causal graphs. We only generate the summary graph to get in line with the other benchmark methods that only support summary graphs. Due to page limit, we only added the quantitative results for inferred temporal graph in the Appendix Section A.4.4.

---

> ### Author Response · Authors · 2022-11-17
> **Point-by-point Response to Reviewer SHw1 (Part 2/2)**
>
>
> > Please add some comments about the difference between Granger causality and ANMs or PCMCI that also identify the temporal causal graph.
>
> This is indeed an important question. We have revised our related works section accordingly.
>
> ANM-based approaches infer causal graph based on Additive Noise Model (e.g. TiMINo [1]), which often does not necessarily hold on real data and the extension to large scale multivariate datasets is challenging. On the contrary, our inference is based a neural network and does not depend on the noise model heavily, so is of higher robustness and can be extended easier.
>
> For conditional independence tests such as PCMCI, the results highly depend on conditional independence tests, which are difficult to implement fast and reliably. Differently, our Granger-causality-based approach models the dynamic process explicitly with powerful Neural Networks and provides convergence guarantee with fair assumptions, and thus permitting joint causal discovery and data imputation.
>
> Benefiting from the above advantageous over pervious methods, our approach demonstrates superior performance, especially when with a large percentage of missing data points.
>
> [1] Mooij, J. M., Janzing, D., Heskes, T., & Schölkopf, B. (2011). On Causal Discovery with Cyclic Additive Noise Models. Advances in Neural Information Processing Systems, 24.
>
>
> > It is unclear how $\tau_{max}$ is chosen. Is this assumed to be known? What if this isn't known?
>
> Here $\tau_{max}$ is unknown. Because our approach is robust to hyperparameters settings and can give decent results across a wide range of $\tau_{max} (3\sim 9)$ (shown in section A.4.6), so we provide a rough estimation in implementation. Specifically, one can choose the largest value of $\tau_{max}$ when significant correlation exists between $X_t$ and $X_{t-\tau_{max}}$. We clarified this issue in section A.4.6 of the revised manuscript.
>
> > All experiments use missing data. It would be great to see a baseline comparing to datasets with full observability.
>
> The comparison results on fully observed time-series are listed in section A.4.5 (Page 17), namely "Causal Discovery with Structured Time-series Data". In these settings our approach beats baselines with a clear margin and the discovery results are near perfect on Lorenz-96 and VAR datasets.
>
> > What's the intuition of using the moving average as a training signal for the imputation network?
>
> The intuition behind the moving average (I suppose you mean Eq. 9 if I understand correctly) is to leverage the delayed supervision from the imputed data points. Specifically, in the early stage of training, the network tends to gives unsatisfatory imputation and thus hampers discovering the underlying causal graph, while introducing moving average can avoid such results effectively.
>
> > For the graph discovery stage - do you calculate an expectation over multiple graph samples or is this amortised over different batches? Or do you use the same graph sample for optimising this loss?
>
> We generate a graph for each training sample according to $\sigma(\theta_{\tau,ij})$, and use Gumbel softmax to optimize the loss. We clarified this problem in section 4.1 of the revised manuscript.
>
> > Eq 3: what is $e$? What's the assumption about it? This might make or break the use of the L2 loss.
>
> There is no $e$ in Eq. 3. Do you mean $\delta$?
> $\delta(\cdot)$ is an indicator function (i.e., $\delta(0)=1$ and $\delta(t)=0, t \neq 0$) serving as observation mask in our formulation. This does not affect the use of L2 loss.
>
> > The model choices about noise distributions and loss functions should be better justified.
>
> Thanks for your suggestions. For noise distributions we use additive noise model, this is a commonly used assumption in causal discovery are proven to works well in experiments section. In simulation experiments we use the standard VAR and Lorenz-96 datasets, which is also simulated with Additive Gaussian Noise.
>
> As for the loss function, the justification is provided in the proof of Theorem 1 (Section A.1.1). This design of loss function allow us to guarantee convergence towards true causal graphs.
>
> > Other miscs
>
> Thank you for the suggestions, and we have amended the manuscript accordingly.
>
> Please let us know if more improvements can be made. Many thanks.

---

> > ### Comment · Reviewer_SHw1 · 2022-12-12
> > **Thanks for the clarifications**
> >
> > Thanks a lot for the thorough rebuttal and clarifications. Most of my questions have been addressed.
> >
> > However, a few questions remain:
> >
> > > > Eq 3: what is ? What's the assumption about it? This might make or break the use of the L2 loss.
> > >
> > > There is no  in Eq. 3. Do you mean ?  is an indicator function (i.e.,  and ) serving as observation mask in our formulation. This does not affect the use of L2 loss.
> >
> > I actually meant e in eq. 1 or 10. (The noise term.) Currently, it seems that this is assumed to be $e \sim \mathcal{N}(0,\sigma)$ with $\sigma$ being the same for all variables to justify an L2 loss. This should be explicitly stated as an assumption. [1, 2] touch upon related problems. It's mentioned in A.1.1 but would be useful in the main text as well.
> >
> > > > The model choices about noise distributions and loss functions should be better justified.
> > >
> > > Thanks for your suggestions. For noise distributions we use additive noise model, this is a commonly used assumption in causal discovery are proven to works well in experiments section. In simulation experiments we use the standard VAR and Lorenz-96 datasets, which is also simulated with Additive Gaussian Noise.
> > > As for the loss function, the justification is provided in the proof of Theorem 1 (Section A.1.1). This design of loss function allow us to guarantee convergence towards true causal graphs.
> >
> > I agree that the experimental section shows good performance of the proposed method. However, the theoretical justification lacks some rigour. E.g. statements such as "Assuming that f(·) accurately models causal relations" are not clearly defined. Furthermore the theorem still depends on a properly chosen $\lambda$ which is only empirically analysed in table 12. However, this analysis is not clear.
> >
> >
> > Lastly, the paper seems to misrepresent results in the results tables - results should not be boldened if they are not statistically significant to another related entry, e.g. table 2 - CUTS vs NGC+ZOH.
> >
> >
> > [1] Kaiser, Marcus, and Maksim Sipos. "Unsuitability of NOTEARS for Causal Graph Discovery when Dealing with Dimensional Quantities." Neural Processing Letters (2022): 1-9.
> > [2] Reisach, Alexander, Christof Seiler, and Sebastian Weichwald. "Beware of the simulated dag! causal discovery benchmarks may be easy to game." Advances in Neural Information Processing Systems 34 (2021): 27772-27784.

---

> > > ### Author Response · Authors · 2022-12-13
> > > **Reply**
> > >
> > > Thank you for your response. We would like to clarify these questions in the following and would be glad to comment on any more questions. However, the discussion stage is about to end. Could you please make responses sooner? Thanks a lot.
> > >
> > > > ...This should be explicitly stated as an assumption. [1, 2] touch upon related problems. ...
> > >
> > > Thank you for the clarification. Indeed $\sigma$ being the same or not for all variables is an important problem. We will add this assumption in the main text in the final version.
> > >
> > > > However, the theoretical justification lacks some rigour. E.g. statements such as "Assuming that $f(\cdot)$ accurately models causal relations" are not clearly defined.
> > >
> > > Thank you for this valuable suggestion. This statement is explained as assumption 1 of Theorem 1 that "DSGNN $f_{\phi_i}$ in Latent data prediction stage model generative function $f_i$ with an error smaller than arbitrarily small value $e_{\text{NN},i}$". We do agree that in the proof the statement "accurately models" is not rigorous enough. We will amend this part in the final version.
> > >
> > > > Furthermore the theorem still depends on a properly chosen $\lambda$ which is only empirically analyzed in table 12. However, this analysis is not clear.
> > >
> > > The question of $\lambda$ is explained with assumption 2 in Theorem 1, which is intuitively explained in Section 4.4. We would like to clarify that selecting $\lambda$ is trivial if all the causal relations are significant enough, e.g. if we assume that
> > > $\forall i,j=1,...,N,  \|f_{\phi_j}(X\odot S_{\tau,ij=1}) - f_{\phi_j}(X\odot S_{\tau,ij=0})\|_2^2 > 0.1$
> > > then selecting any $\lambda < 0.1p$ ($p$ is the missing prorbability) would allow us to guarantee convergence. Experiments in table 12 is to empirically confirm that selecting proper $\lambda$ is trivial.
> > >
> > > >  ...results should not be boldened if they are not statistically significant to another related entry...
> > >
> > > Thank you for this valuable suggestion. We will unbold some numbers without statistical significance in the final version.
> > >
> > > Thanks again for your response! This will allow us to further improve our paper.

---

> > > > ### Comment · Reviewer_SHw1 · 2022-12-13
> > > > **Response**
> > > >
> > > > Thanks for the quick response. I have updated my score to 6.
> > > >
> > > > Some comments below:
> > > >
> > > > > > ...results should not be boldened if they are not statistically significant to another related entry...
> > > > >
> > > > > Thank you for this valuable suggestion. We will unbold some numbers without statistical significance in the final version.
> > > >
> > > > Another option would be to bolden all best performing methods
> > > >
> > > > > > Furthermore the theorem still depends on a properly chosen  which is only empirically analyzed in table 12. However, this analysis is not clear.
> > > > >
> > > > > The question of $\lambda$ is explained with assumption 2 in Theorem 1, which is intuitively explained in Section 4.4. We would like to clarify that selecting $\lambda$ is trivial if all the causal relations are significant enough, e.g. if we assume that then selecting any  ( is the missing prorbability) would allow us to guarantee convergence. Experiments in table 12 is to empirically confirm that selecting proper  is trivial.
> > > >
> > > > I would urge the authors to be more specific and rigorous when it comes to the theoretic analysis of the work. In its current form, I find the paper to mainly present convincing experimental results but find the theoretical aspects lacking.
> > > > E.g., statements like `that selecting $\lambda$ is trivial if all the causal relations are significant enough` are very handwavy. When is a causal relation "significant enough"? As such I am happy to increase my score to 6 based on some of the clarifications but would recommend refining the theoretical aspect of the paper.

---

> > > > > ### Author Response · Authors · 2022-12-13
> > > > > **Thank you so much!**
> > > > >
> > > > > Your comment is of great help to us. Thank you so much.
> > > > >
> > > > > Indeed some terms in the reply we just wrote are relatively intuitive since the rebuttal phase is ending soon and we have to write in a short time. We apologize for that.
> > > > >
> > > > > > When is a causal relation "significant enough"?
> > > > >
> > > > > Sorry for the confusion about the word "significant" in the reply. What we are really trying to say is that when $\forall i,j=1,...,N,  \|f_{\phi_j}(X\odot S_{\tau,ij=1}) - f_{\phi_j}(X\odot S_{\tau,ij=0})\|_2^2 > \lambda_0$, then selecting any $\lambda < p\lambda_0$ will ensure convergence.
> > > > >
> > > > > We will make a more rigorous refinement on the theoretical aspects of this paper (e.g. section A1.1) in the next few days. We are quite sure that these expressions will be much better by then.

---

### Official Review · Reviewer_fTfw · 2022-10-30

**Confidence:** 4
**Correctness:** 3
**Technical Novelty And Significance:** 2
**Empirical Novelty And Significance:** 3
**Recommendation:** 8

**Clarity, Quality, Novelty And Reproducibility:**

### Clarity

The paper could clearly improve on clarity as I wrote above. I think the idea is good and that it would be possible to convey it in a more straightforward fashion to the community. I pointed to key points that should be addressed but I encourage the authors to work on the presentation.

### Quality

The authors provide several experimental results and perform many ablation studies to study the impact of each part of their model. I pointed above to additional experiments that would help the reader understand more deeply the behaviour of this model.

### Novelty

While Granger causality for time series is very well established, the contribution of this work relies on the adaptive training scheme (which further justifies why experiments with varying number of refinement stages are needed).

### Reproducibility

Reproducibility is inherently linked to clarity, which can be improved.

**Details Of Ethics Concerns:**

I have no ethical concerns.

**Strength And Weaknesses:**

### Strengths

- This paper addresses an important problem in causal discovery : inferring causal dependencies from irregular time series.
- The method appears to outperform baselines it is comparing against.
- The iterative refinement of the imputation is a good idea and the authors show that the data fitting and causal graph discovery are helping each other.
- The paper proposes different ablations (though I believe complementary experiments would be required - see below).


### Weaknesses


- The biggest weakness of this paper is the clarity. I think Figure 1 and Figure 2 should be mixed together so you make very clear that each stage 1-2-3 corresponds to doing one iteration of Figure 1. The notation at the beginning of Section 4.2 is uncommon as well (what is $p(x \rightarrow p(x))$, this lacks rigour. Similarly, $\theta$ is not directly understood as parametrising $m$. Only in point of Theorem 1 does it become apparent, which is, I believe, the wrong place to introduce this concept.

- In the definition of Granger causality, I think there is a mistake in the causal direction. If I’m not mistaken, you mean $j$ causes $i$.

- The implications of the assumptions of Theorem 1 should be discussed in the main text. It’s important for the reader to have an intuitive grasp of when this theorem holds.

- It seems that each stage is a refinement of the causal graph learning by using more accurate imputations. In that respect, I think it would make sense to have an experiment that investigates the impact of a different number of refinement steps. It does not seem to be present in the ablation study section.

- The related section misses the whole line of work of causality in time series using convergent cross mapping. I encourage the authors to show how their paper situates with respect to that literature.  In particular, extensions of such methods have been proposed for irregular time series (e.g. Latent Convergent Cross Mapping - https://openreview.net/pdf?id=4TSiOTkKe5P), which represents a relevant baseline for the problem discussed here.

- You are comparing your approach with a GP for imputation. I believe a GP is non causal as it may use observations from the future to predict information from the past. I have few questions regarding this. First do you think it is an issue that the imputation is non-causal. Second, if not, would your approach fare better if another non-causal imputation strategy would be used (e.g. BRITS that you are referring to). In which case, a single step of imputation followed by a causal graph learning step could be enough to reach similar performance.

- I think unstructured time series is not the appropriate name for the type of time series you are considering. In my experience, this is more often referred to as *irregular* time series.

- Despite the bold numbers, it seems difficult to appreciate the magnitude of the improvement in performance this method is giving. Indeed, the difference between AUC results are very small.

- Minor : Right above equation 11, I believe you mean last $n_3$ Epochs and not $n_2$.

**Summary Of The Paper:**

This paper proposes an architecture to infer causal link from irregularly sampled time series data. The authors embrace the Granger causality paradigm. The method works by interleaving missing data imputation scheme and causal graph learning. The causal graph is used to structure the imputation predictions by using only causal predictors. The inferred causal graph is assumed to get more accurate as the imputation gets better. The model proceeds in three stages. The first is learning the forecasting / imputation and causal graph using the initial imputation. The second is similar but replaces the missing values with the predictions from stage 1. The third further refines the causal graph learning by replacing the missing values by the imputation from the second stage. The causal graph of the last stage is the predicted causal graph of the model.

**Summary Of The Review:**

Interesting idea that would deserve additional experiments to fully grasp the contribution of the method. The clarity and the positioning with respect to existing works could be improved.

---

> ### Author Response · Authors · 2022-11-17
> **Point-by-point Response to Reviewer fTfw (Part 1/2)**
>
>
> Thank you for taking time to review this work. Your suggestions are of great help to improve this manuscript. We would like to clarify many important points in the following and amend our paper accordingly.
>
> > I think Figure 1 and Figure 2 should be mixed together so you make very clear that each stage 1-2-3 corresponds to doing one iteration of Figure 1.
>
> We agree that mixing Fig. 1 and Fig. 2 can present better and have updated the manuscript according to your valuable suggestion.
>
> > The notation at the beginning of Section 4.2 is uncommon as well (what is $p(x\rightarrow p(x))$, this lacks rigour. Similarly, $\theta$ is not directly understood as parametrising $m$. Only in point of Theorem 1 does it become apparent, which is, I believe, the wrong place to introduce this concept.
>
> Thanks a lot for the constructive comment. In the revised version, we have changed $p(x_{t-\tau,i}\rightarrow p(x_{t,j}))$ to $p(x_{t-\tau,i}\rightarrow x_{t,j})$ and explicitly explained the parameters $m$ and $\theta$ at the beginning of Section 4.2.
>
> > In the definition of Granger causality, I think there is a mistake in the causal direction. If I’m not mistaken, you mean $j$ causes $i$.
>
> Many thanks for your careful reading and we have corrected this mistake now.
>
> > The implications of the assumptions of Theorem 1 should be discussed in the main text. It’s important for the reader to have an intuitive grasp of when this theorem holds.
>
> Thank you for this valuable suggestion! The intuitive implications of these assumptions are as follows:
>
> Assumption #1 is intrinsically the Universal Approximation Theorem [1] of neural networks, i.e., the network is of an appropriate structure and fed with sufficient training data.
>
> Assumption #2 means there exists a threshold $\lambda_{0}$ to binarize $\| f_{\phi_i}(X\odot S_{\tau,ij=1}) - f_{\phi_i}(X\odot S_{\tau,ij=0}) \|$, serving as an indicator as to whether time-series $j$ contributes to prediction of $i$.
>
> [1] Hornik, K., Stinchcombe, M., & White, H. (1989). Multilayer feedforward networks are universal approximators. Neural Networks, 2(5), 359–366.
>
> > It seems that each stage is a refinement of the causal graph learning by using more accurate imputations. In that respect, I think it would make sense to have an experiment that investigates the impact of a different number of refinement steps. It does not seem to be present in the ablation study section.
>
> Thank you for your advice. We added an experiment to reveal the evolution of performance with increasing learning steps. The results can be found in Appendix Section A.4.3.
>
> > The related section misses the whole line of work of causality in time series using convergent cross mapping. I encourage the authors to show how their paper situates with respect to that literature. In particular, extensions of such methods have been proposed for irregular time series (e.g. Latent Convergent Cross Mapping - <https://openreview.net/pdf?id=4TSiOTkKe5P>), which represents a relevant baseline for the problem discussed here.
>
> We totally agree that this line of works are closely related to our work, especially Latent Convergent Cross Mapping (LCCM) specially designed for handling irregular sequences, and cited these works [1][2][3][4] in the revised version.
>
> LCCM can serve as a good benchmark for causal discovery with irregular data, and we added performance comparison on multiple irregular time-series datasets in the new experiment section. We found that LCCM performs greatly under some settings but is unsatisfactory in other scenarios (e.g. VAR datasets simulated without ODE). On the contrary, Our approach perform well in most settings and the improvements are especially large with a large percentage of missing values.
>
> [1] Benkő, Z., Zlatniczki, Á., Stippinger, M., Fabó, D., Sólyom, A., Erőss, L., Telcs, A., & Somogyvári, Z. (2020). Complete Inference of Causal Relations between Dynamical Systems (arXiv:1808.10806). arXiv.
>
> [2] Sugihara, G., May, R., Ye, H., Hsieh, C., Deyle, E., Fogarty, M., & Munch, S. (2012). Detecting Causality in Complex Ecosystems. Science, 338(6106), 496–500.
>
> [3] Brouwer, E. D., Arany, A., Simm, J., & Moreau, Y. (2021, March 17). Latent Convergent Cross Mapping. International Conference on Learning Representations.
>
> [4] Ye, H., Deyle, E. R., Gilarranz, L. J., & Sugihara, G. (2015). Distinguishing time-delayed causal interactions using convergent cross mapping. Scientific Reports, 5(1), Article 1.

---

> ### Author Response · Authors · 2022-11-17
> **Point-by-point Response to Reviewer fTfw (Part 2/2)**
>
>
>
> > You are comparing your approach with a GP for imputation...
>
> This is a quite interesting question, thank you for this valuable insight. In our opinion, the non-causal imputation is not appropriate for causal discovery for the following two reasons
>
>  1.  The non-causal imputation would hamper the causal discovery process, since time series intuitively are Markov-like and it is unreasonable to predict backward.
>
>  2.  Non-causal imputation approaches are prone to overfitting to the observations, so would degenerate at a large missing percentage. Conversely, we can reach similar performance by simply conducting imputation and causal graph learning sequentially when with few entries. This is well supported by experimental results in Figure 4.
>
> > I think unstructured time series is not the appropriate name for the type of time series you are considering. In my experience, this is more often referred to as irregular time series.
>
> Many thanks for your kind suggestion. We changed the term "unstructured" into "irregular" through the paper.
>
> > Despite the bold numbers, it seems difficult to appreciate the magnitude of the improvement in performance this method is giving. Indeed, the difference between AUC results are very small.
>
> We respectfully disagree with the judgment and argue that: (i) when the data are with more missing data, our approach largely raises AUC; (ii) at a low percentage of missing entries, our approach provides slight improvement but near perfect results.  The details figures are as follows:
> 1. The results on 6/10 settings with a large percentage of missing data, where the increment compared to the second competitor as
> | Setting| Increments of AUC |
> |:-----------------:|:-----------:|
> | VAR with $p=0.6$ | **2.83%** |
> | VAR with $T_{max}=4$ | **1.74%** |
> | Lorenz-96 with $p=0.6$ | **1.13%** |
> | Lorenz-96 with $T_{max}=4$ | **1.47%** |
> | Netsim with $p=0.1$ | **3.81%** |
> | NetSim with $p=0.2$ | **0.40%** |
> 2. The improvement of AUC score on 4/10 cases with few missing entries (VAR and Lorenz-96 with $p=0.3$ and $T_{max}=2$) is indeed relatively small, however, the AUC is very close to 1 (e.g. 0.9996 for Lorenz-96 with $p=0.3$) and even small improvement on such a high AUC is challenging and a big step towards perfection.
>
> > Minor : Right above equation 11, I believe you mean last $n_3$ Epochs and not $n_2$.
>
> Thank you for your correction. We have corrected this notation in the revised manuscript.

---

> ### Comment · Reviewer_fTfw · 2022-12-06
> **Thank you for your responses**
>
> Thank you for carefully addressing all my comments. I have no more questions and have updated my score accordingly.
>
> Best Regards

---

> > ### Author Response · Authors · 2022-12-07
> > **Thanks!**
> >
> > Thanks for your reply and appreciation! This means a lot to us.

---

### Author Response · Authors · 2022-11-17
**Official Response to All Reviewers**

## 13 Dec Update

Since the discussion stage is closing, we would like to thank all the reviewers for taking much effort in reviewing our paper. Moreover, during the rebuttal phase, they actively replied to our responses. Their efforts really help to improve this work. Special thanks to them!

As reviewer SHw1 pointed out, a few theoretical terms in the paper may be confusing. So we are making a more rigorous refinement on the theoretical expressions of this paper (e.g. section A1.1 and Theorem 1) for the (possible) final version.

---

## 18 Nov Official Response

We deeply thank the reviewers for their insightful feedback. These suggestions greatly help to improve our work. In the following discussions, we individually address the reviewers’ comments.

We would like to clarify our contribution and novelty to all reviewers here:
1. We are the first to introduce joint data imputation into irregular/unstructured time-series causal discovery problem [1][2], which is conducted in an iterative optimization scheme and provides superior performance.
2. We use explicit functional dynamic modeling of causal effects and guarantee convergence given appropriate assumptions, which are substantially different from existing variational-based joint optimization methods (e.g., [3][4]) that only apply to static settings and have no convergence guarantee.
3. Besides data imputation and causal discovery, combining neural networks and Granger causality would also benefit the explanability and robustness of other diverse downstream tasks (feature selection, prediction, classification, etc.).

We briefly list the manuscript revisions as follows:
1. Update the related work section to include relevant works mentioned by the reviewers.
2. Update proof of Theorem 1 to include more details about algorithm convergence and further analyze how data imputation boosts causal discovery.
3. Add several experiments (including validation on real data and ablation studies) to address the reviewers' concerns and support our contribution.
4. Address some minor issues.

We hope our revised manuscript and additional experiments help to provide further clarity. Please let us know if any improvement can be made.

[1] Bellot, A., Branson, K., & Schaar, M. van der. (2022, February 3). Neural graphical modelling in continuous-time: Consistency guarantees and algorithms. International Conference on Learning Representations.

[2] Brouwer, E. D., Arany, A., Simm, J., & Moreau, Y. (2021, March 17). Latent Convergent Cross Mapping. International Conference on Learning Representations.

[3] Morales-Alvarez, P., Gong, W., Lamb, A., Woodhead, S., Jones, S. P., Pawlowski, N., Allamanis, M., & Zhang, C. (2022). Simultaneous Missing Value Imputation and Structure Learning with Groups (arXiv:2110.08223). arXiv.

[4] Geffner, T., Antoran, J., Foster, A., Gong, W., Ma, C., Kiciman, E., Sharma, A., Lamb, A., Kukla, M., Pawlowski, N., Allamanis, M., & Zhang, C. (2022). Deep End-to-end Causal Inference (arXiv:2202.02195). arXiv.

---

### Public Comment · ~B_L1 · 2023-03-14
**Is CUTS directly applied for irregular time series with data missing initially?**

Hello authors, it's my honor to ask you for my confusion. In my opinion, the initial time seires data CUTS uses are complete at every time point.  I wonder if CUTS could directly use irregular time series that have missed some data when got.
I appreciate authors's response. Thanks.

---

> ### Author Response · Authors · 2023-03-14
> **We Initially Fill the Missing Data with ZOH (Section A.3.2)**
>
> Thank you for your question. Indeed, we initially fill the missing data with ZOH (see Section A.3.2). As a result, the initial time-series data CUTS uses are somewhat "complete" at every time point. For any irregular time series, we can always perform uniform sampling and fill the unobserved time points with an initial guess (such as ZOH).
>
> As for direct input of irregular time series, CUTS currently is not designed to do that. However, this is an interesting topic for future work.
>
> Thanks again.

---

### Decision · Program_Chairs · 2023-01-20

**Decision:**

Accept: poster

**Justification For Why Not Higher Score:**

 Issues regarding clarity and some reviewers were not fully convinced by the experiments.

**Justification For Why Not Lower Score:**

The main concern of the most negative reviewer regarding technical rigor to support some claims that data imputation supporting causal discovery were addressed well in the rebuttal.

**Metareview: Summary, Strengths And Weaknesses:**

The paper presents a model for joint causal graph learning and missing data imputation for unstructured temporal data. The model is trained in an EM-style manner. The paper compares the method to relevant temporal discovery methods and achieves competitive performance.

Strengths + Weakneses: The paper offers interesting ideas to an important problem and outperforms current SOTA methods. Experiments were considered very convincing and the general attitude of the reviewers towards this paper is positive. One reviewer had concerns with the technical rigor to support some claims on data imputation supporting causal discovery. Yet, this concern was addressed by the rebuttal adequately (unforutnately the reviewer was unresponsive to that). I recommend acceptance of the paper.

**Note From Pc:**

if the above contains the word "oral" or "spotlight" please see: "oral" presentation means -> notable-top-5% and "spotlight" means -> notable-top-25%. As stated in our emails, we are disassociating presentation type from AC recommendations

**Summary Of Ac-Reviewer Meeting:**

N/A